# Hybrid AI-assistive diagnostic model permits rapid TBS classification of cervical liquid-based thin-layer cell smears

Xiaohui Zhu[1,2,16], Xiaoming Li[3,16], Kokhaur Ong[4,15,16], Wenli Zhang[1,2,16], Wencai Li[5,16], Longjie Li[4], David Young[4], Yongjian Su[6], Bin Shang[6], Linggan Peng[6], Wei Xiong[7], Yunke Liu[8], Wenting Liao[9], Jingjing Xu[5], Feifei Wang[1,2], Qing Liao[1,2], Shengnan Li[6], Minmin Liao[1,2], Yu Li[1,2], Linshang Rao[6], Jinquan Lin[6], Jianyuan Shi[6], Zejun You[6], Wenlong Zhong[10], Xinrong Liang[10], Hao Han[4], Yan Zhang[1,11], Na Tang[12], Aixia Hu[13], Hongyi Gao[14], Zhiqiang Cheng[12✉], Li Liang[1,2✉], Weimiao Yu[4,15✉] & Yanqing Ding[1,2✉]

Technical advancements significantly improve earlier diagnosis of cervical cancer, but accurate diagnosis is still difficult due to various factors. We develop an artificial intelligence assistive diagnostic solution, AIATBS, to improve cervical liquid-based thin-layer cell smear diagnosis according to clinical TBS criteria. We train AIATBS with >81,000 retrospective samples. It integrates YOLOv3 for target detection, Xception and Patch-based models to boost target classification, and U-net for nucleus segmentation. We integrate XGBoost and a logical decision tree with these models to optimize the parameters given by the learning process, and we develop a complete cervical liquid-based cytology smear TBS diagnostic system which also includes a quality control solution. We validate the optimized system with >34,000 multicenter prospective samples and achieve better sensitivity compared to senior cytologists, yet retain high specificity while achieving a speed of <180s/slide. Our system is adaptive to sample preparation using different standards, staining protocols and scanners.

[1] Department of Pathology, Nanfang Hospital and Basic Medical College, Southern Medical University, Guangzhou, Guangdong Province, PR China. [2] Guangdong Province Key Laboratory of Molecular Tumor Pathology, Guangzhou, Guangdong Province, PR China. [3] Department of Pathology, Shenzhen Bao'an People's Hospital (group), Shenzhen, Guangdong Province, PR China. [4] Institute of Molecular and Cell Biology, A*STAR, Singapore, Singapore. [5] The First Affiliated Hospital of Zhengzhou University, Zhengzhou, Henan Province, PR China. [6] Guangzhou F.Q.PATHOTECH Co., Ltd, Guangzhou, Guangdong Province, PR China. [7] Guangzhou Kaipu Biotechnology Co., Ltd, Guangzhou, Guangdong Province, PR China. [8] Laboratory Department, Guangzhou Tianhe District Maternal and Child Health Care Hospital, Guangzhou, Guangdong Province, PR China. [9] State Key Laboratory of Oncology in South China, Collaborative Innovation Center for Cancer Medicine, Sun Yat-sen University Cancer Center, Guangzhou, Guangdong Province, PR China. [10] Guangzhou Huayin medical inspection center Co., Ltd, Guangzhou, Guangdong Province, PR China. [11] Department of Pathology, Shenzhen Longhua District Maternity & Child Healthcare Hospital, Shenzhen, PR China. [12] Department of Pathology, Shenzhen First People's Hospital, Shenzhen, Guangdong Province, PR China. [13] Department of Pathology, Henan Provincial People's Hospital, Zhengzhou, Henan Province, PR China. [14] Department of Pathology, Guangdong Provincial Women's and Children's Dispensary, Shenzhen, Guangdong Province, PR China. [15] Bioinformatics Institute, A*STAR, Singapore, Singapore. [16]These authors contributed equally: Xiaohui Zhu, Xiaoming Li, Kokhaur Ong, Wenli Zhang, Wencai Li. ✉email: chengzhiqiang2004@aliyun.com; lli@fimmu.com; yu_weimiao@bii.a-star.edu.sg; dyqgz@126.com

Cervical cancer (CC) is one of the most common malignant tumors in women, with 604,000 new cases and 342,000 deaths in 2020[1]. In developing countries, the mortality rate of CC is about 2.5 times than that of developed countries. One of the reasons is that CC screening has been successfully carried out in developed countries, and early diagnosis enables close monitoring of disease progression and timely intervention and treatment[2]. Cytology screening technology has also evolved from the traditional Pap smear to liquid-based thin-layer cell smear technology, which has significantly improved the quality of samples and is commonly used for investigation in CC screening[3]. However, the incidence and mortality of CC in China is increasing with an annual growth rate of 2–3%[4]. We also observed that younger women are developing this disease[4]. The increasing trend of CC in China, probably in other developing countries too, is due to a number of complex factors, but the shortage of experienced cytologists and cytotechnologists is one of the major factors.

In addition to specimen preparation techniques, diagnostic criteria are essential for patient stratification. The Bethesda system, known as the TBS standard, was established in 1988 and has attracted the global attention of cytopathologists. Compared with the traditional five-level classification system, the current TBS standard (2014 version)[5], as shown in Supplementary Table 01 provides more clinical guidance. Currently, TBS is a widely accepted and adopted diagnostic standard in different countries, including China. However, the accurate diagnosis of cervical liquid-based cytology smears is still a challenge due to the following two reasons: (I) technically speaking, a cytologist must find only a few abnormal and malignant cells in a sample composed of tens of thousands of cells, (II) in addition, due to the lack of experienced and qualified cytologists or cytotechnologists, and the influence of factors such as their diagnostic experiences, moods, fatigue, etc. human factors may cause data misinterpretation.

The emergence of AI in the 21st century has shown great promise to perform such crucial but tedious tasks both thoroughly and tirelessly. AI has changed our daily lives in various ways, and its applications in medical diagnosis has increased rapidly. The field of AI has been well developed, including reinforcement learning, supervised learning and unsupervised learning, such as machine learning (ML), pattern recognition, convolutional neural networks (CNNs), feedback neural networks, self-supervised learning, and weakly supervised learning, etc. Today, these technologies have an increasingly important role in biomedical and clinical application[6]. Different AI systems have been developed to address the needs in clinical diagnosis, for example cardiac systolic dysfunction screening[7] and skin cancer classification which achieved diagnostic results that are comparable to the level of clinical experts[8].

In pathological images, the development of digital pathology and big data facilitate the development of traditional image analysis and ML technology[9]. At the same time, it is especially worth noting that deep learning (DL) also has been extensively studied in pathology, and CNNs have become the preferred technology for general image classification[10]. It also has been used for image-based detection tasks[11,12] and applied to identify and quantify cellular[13] and histological features[14,15].

In clinical pathological diagnosis, the application of AI to whole-slide imaging (WSI) pathological image detection, classification and prognosis prediction has been a hot research topic in recent years. Various studies were published, including the development of CNNs to classify pathological breast cancer slice images at the pixel level[16]. Neighboring Ensemble Predictor neural network, based on CNNs, was proposed to detect nuclei and classify them using spatial constraints[17]. For the prediction of cancer diagnosis and progression/prognosis, we can use AI to diagnose basal cell carcinoma, prostate cancer, and breast cancer axillary lymph node metastasis through weakly supervised learning[18], classify lung cancer samples and make prognosis based on gene mutation profiles[19], and predict the prognosis of colorectal cancer patients[20]. These research results have laid a solid foundation for the clinical application of AI in pathological diagnosis and prognosis.

AI-assistive CC screening is the most widely used application in pathology[21] and has a wide range of application prospects[22]. Since the birth of cell smear technology, people have tried to develop a system that can automatically screen cervical cell samples. The Cytoanalyzer project, which can be traced back to the 1950s in the United States, developed an automatic microscope for differentiating cancer versus normal cells based on the size and optical density of the nucleus for the screening of Pap smears[23]. Japan's CYBEST system attempts to extract features such as nuclear area, nuclear density, cytoplasmic area, and nuclear-to-plasma ratio from traditional cell images;[24] Neuromedical Systems' PAPNET cytology screening system began to use neural network classifiers to identify abnormal cells and automatically screen cervical smears[25]. In recent years, the development of neural network technology has led to the application of computer-assisted cervical cytology screening to receive increasing attention. For example, image classification of cervical Pap smears[26–29] and target detection and classification of cervical Pap smears[30] have been implemented. Commercialized products have even been developed and approved by the U.S. Food and Drug Administration[31]. At the same time, there are also many reports of clinical research using AI-assisted analysis systems[32–36]. Without exception, these systems have increased the sensitivity of CC screening and reduced the false negative rate. However, many studies only focused on a certain part of the automatic screening diagnosis of cervical Pap smear, such as using six different CNNs to classify cervical lesions[37], or using different algorithms to segment cervical cell and nuclei[38] and detect and classify images of PAP smears[39]. A variety of automatic screening systems are able to identify suspicious intraepithelial lesion areas[31,40] and allow doctors to focus on those suspicious areas.

Currently, there is no such system for automated assistive interpretation of cervical liquid-based cytology smears in strict accordance with TBS standards which includes all the subtyping (squamous, glandular, and infectious lesions). The challenges of building a robust and adoptable AI-assistive system are the potential variations and factors caused by (I) different sediment approaches, such as natural, membrane, and centrifugal sedimentations, (II) staining reagents, including both commonly used EA-50 and EA-36, (III) image quality variation caused by different scanners, and (IV) sample preparation at multiple medical centers. Large-scale study and systematic development are required to address these challenges such that the developed AI model is able to be adopted by different medical centers. The desired diagnostic system will require a powerful and robust AI-based model that is easily applied, verified and adopted by cytopathologists and widely applicable to different populations and multiple medical centers. It will reduce the workload of cytologists, reduce false negative rates, improve diagnostic accuracy, and ultimately reduce mortality. This urgent and unmet need is especially important to reduce the rising mortality rate in China and other countries facing similar challenges.

In this paper, we have developed an Artificial Intelligence (AI) assistive diagnostic system that can help cytologists perform data interpretation and improve the efficiency and quality of screening based on TBS standards. The integrated system is composed of five AI models, which we employ to detect and classify the lesions

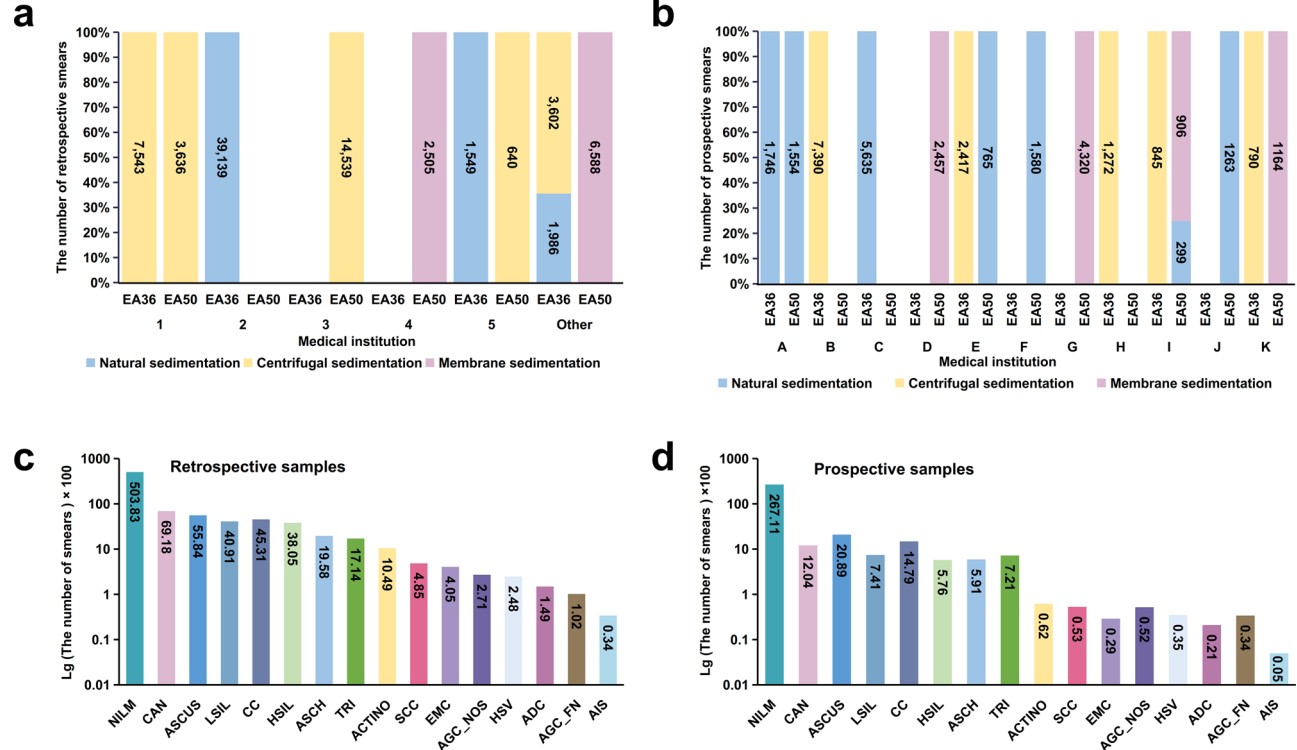

**Fig. 1 Large-scale multicenter smear samples collection based on the TBS standard. a** The number of retrospective smears of different preparation and staining protocols collected from multiple medical institutions ($n = 81,727$). **b** The number of prospective smears of different preparation and staining protocols collected from multiple medical institutions ($n = 34,403$). **c** The smear number of different TBS classification in the retrospective study. **d** The smear number of different TBS classification in the prospective study.

based on large and annotated training datasets collected from five central medical centers in China. Based on a large, well-organized training database, we developed an AI-Assistive TBS (AIATBS) to detect and classify digital cervical liquid-based cytology smears. Adhering to the requirements of the TBS reporting system, we can classify cervical precancerous lesions and infectious lesions with sensitivity exceeding that of senior cytologists, while specificity was slightly lower but comparable. The clinical validation at 11 medical centers, combined with the sensitivity and specificity profile, shows that AIATBS system is suitable for adoption by clinical centers for routine assistive diagnostic screening.

## Results

**Large-scale smear sample collection and digitization for multicenter retrospective and prospective studies.** In order to develop the AIATBS system, we collected a large number of cervical liquid-based thin-layer cell smear samples ($n = 81,727$), from five central medical institutes (indicated by "Medical institutions 1–5" in Fig. 1a) and other anonymized smaller medical institutions (collectively indicated by "Other" in Fig. 1a) within 3 years before May 31, 2019 as our training, testing and validation dataset for the AIATBS system development. The prospective clinical samples ($n = 34,403$) between August 1st, 2019 and April 30th, 2020 from 11 anonymized medical institutions (indicated by "Medical institutions A–K" in Fig. 1b) were collected for our clinical validation.

The smear samples were prepared across the different clinical sites, including use of three common sedimentation methods: (I) natural sediment, for example BD PrepStainTM liquid-based cell automatic film maker, (II) membrane sediment, such as Thinprep liquid-based thin-layer cell preparation system, (III) centrifugal sediment, mostly using centrifugal and manual operation. Cytoplasmic staining reagents included[41] both Papanicolaou

Stain EA-50 Solution (EA-50) and Papanicolaou Stain EA-36 Solution (EA-36), which are routinely used in cervical liquid-based thin-layer cell smear sample staining for clinical screening. The details of the prepared samples using different methods and staining reagents are given in Fig. 1a, b. We can see that there are preferences of different medical centers for sediment and staining methods. After quality control of sample preparation and image acquisition, we scanned the smears and classified each smear/sample according to the TBS diagnosis. The details on final confirmation criteria of smear TBS classification are described in the "Methods". The number of different TBS classifications are as shown in Fig. 1c, d.

**YOLOv3 model achieves fast, semi-supervised smear sample annotation based on the TBS standard.** According to the TBS diagnostic classification, we needed to identify both infected benign and malignant lesions. We classified and annotated both infected and malignment cases and included the benign classes such as superficial squamous epithelium, repair and metaplastic cells, normal glandular epithelial cells, and parabasal cells or lymphocytes, etc. In order to get more detailed pathological features, we further classified the smear samples into 24 new, more detailed classes as show in Table 1 and Fig. 2a. These 24 classes are named as C1–C24 in the classification index of Table 1.

The annotations of a quality-controlled image database are the cornerstone for supervised learning. The fully manual annotation of such a large number of digital pathology images is very challenging, essentially infeasible and unpractical. In order to rapidly select high-efficiency detection algorithms to build our target detection model for semi-supervised annotation, we used our initially available labeled data of squamous intraepithelial lesions to test different algorithms. We found that YOLOv3[42] detection accuracy was comparable with other models, such as

**Table 1 Corresponding relationship between new annotation classification and TBS classification.**

| New annotation classification | Corresponding TBS classification | Category | Definition |
|---|---|---|---|
| AGC_A (C1) | AGC_NOS, AGC_FN, AIS, ADC | Glandular intraepithelial lesions | Atypical glandular cells, pathological morphology similar to cervical gland cells |
| AGC_B (C2) | AGC_NOS, AGC_FN, AIS, ADC | Glandular intraepithelial lesions | Atypical glandular cells, other pathological morphology |
| ASC_L_S (C3) | ASCUS, LSIL, ASCH, HSIL, SCC | Squamous intraepithelial lesion | Single atypical squamous cell with low nuclear/cytoplasmic ratio (excluding Koilocytes) |
| KC (C4) | LSIL | Squamous intraepithelial lesion | Single koilocyte |
| ASC_L_F (C5) | ASCUS, LSIL, ASCH, HSIL, SCC | Squamous intraepithelial lesion | Atypical squamous cells with low nuclear/cytoplasmic ratio, arranged in sheets or clusters |
| ASC_H_B (C6) | ASCUS, ASCH, HSIL, SCC | Squamous intraepithelial lesion | Atypical squamous cells with high nuclear/cytoplasmic ratio, arranged in clusters (cells number $\geq 10$) |
| ASC_H_M (C7) | ASCUS, ASCH, HSIL, SCC | Squamous intraepithelial lesion | Atypical squamous cells with high nuclear/cytoplasmic ratio, arranged in clusters ($2 \leq$ cells number $\leq 10$) |
| ASC_H_S (C8) | ASCUS, ASCH, HSIL, SCC | Squamous intraepithelial lesion | Single atypical squamous cell with high nuclear/cytoplasmic ratio |
| SCC_R (C9) | SCC | Squamous intraepithelial lesion | Keratinizing squamous cell carcinoma cells |
| SCC_G (C10) | SCC | Squamous intraepithelial lesion | Non-Keratinizing squamous carcinoma cells |
| MC (C11) | NIL | Normal | Superficial squamous epithelium |
| SC (C12) | NIL | Normal | Single normal cell with high nuclear/cytoplasmic ratio (Including parabasal cells, lymphocyte cells, and reserve cells etc.) |
| RC (C13) | NIL | Normal | Repair and metaplasia cells |
| GEC (C14) | NIL | Normal | Glandular epithelial cells of cervical tube |
| EMC (C15) | EMC | EMC | Endometrial cells |
| TRI (C16) | TRI | Infectious lesion | Trichomonas vaginalis |
| CAN (C17) | CAN | Infectious lesion | Candida albicans |
| HSV (C18) | HSV | Infectious lesion | Herpes simplex virus |
| ACTINO (C19) | ACTINO | Infectious lesion | Actinomycetes |
| CC (C20) | CC | Infectious lesion | Clue cells |
| PH (C21) | NIL | Evidence of infectious lesion | Perinuclear halo |
| Neutrophils (C22) | NIL | Normal | Neutrophils |
| Mucus (C23) | NIL | Normal | Mucus |
| Debris (C24) | NIL | Normal | Debris |

Faster R-CNN[43], SSD513[44], and RetinaNet[45], as shown in Supplementary Table 02. However, the YOLOv3's efficiency was 2.5–4× higher than others and it was therefore selected to build our single-class detection model. To reduce the labeling burden of cytologists and achieve rapid training data annotation, we used the single-class YOLOv3 model for semi-supervised labeling[46] as mentioned in the Methods, Fig. 2b–f. In total, we annotated *1.7 million* different annotations as a training dataset for our AI learning, as shown in Fig. 2g. We can see that the distribution of different classes is highly unbalanced. For example, AGC_B (C2) and EMC (C15) are still rare cases in our large database.

**Deep learning models can extract high quality features from the training dataset**. After annotation as mentioned above, we trained a multi-class YOLOv3 model to detect intraepithelial lesions, infectious lesions and endometrial cells as detailed in the Methods. The mean Average Precision (mAP) of YOLOv3 model reached 82.33% precision in Fig. 3a, and its detection accuracies of different classes are shown in Fig. 3b. We observed that the accuracy of different classes has some variation. In order to solve such variation and improve the overall performance, we trained an Xception model[47] for fine-grained classification with all annotated and classified images. This model achieved precision of 96.30% and recall of 96.80% as shown in Fig. 3c. We will use these two models to detect and classify targets and extract the

classification and probability information of the target output for further processing.

The targets detected by the YOLOv3 model were reclassified by the Xception model, however there were often false positives that reduced the system performance. Thus, it was important to further reduce the falsely detected targets. We trained the Patch model[48] to classify the 608*608 pixel area (including the target and its surrounding neighborhood). The results shows that the Patch model achieved validation accuracy of 91.92% as shown in Fig. 3d. After the probabilities of true and false Patch classification corresponding to the output targets were analyzed, the patch model could identify the false positives as shown in Supplementary Fig. 1a.

In cytopathological diagnosis, it is crucial to characterize cell nuclei morphology. To address this, SC, ASC_L_S and ASC_H_S data were merged, and a U-Net[49] algorithm was used to train the nucleus segmentation model. The representative segmentations were shown in Fig. 3e, and the mean Intersection Over Union (mIOU) reached 83.60% as in Supplementary Table 03. For prediction, we input the SC, ASC_L_S, and ASC_H_S classified from the Xception model into the cell nucleus segmentation model. The segmented nuclei are shown in Supplementary Fig. 1b, and the gray value of cell nuclei was calculated.

**XGBoost model and logical decision trees can predict TBS classification with high sensitivity and specificity**. In practice,

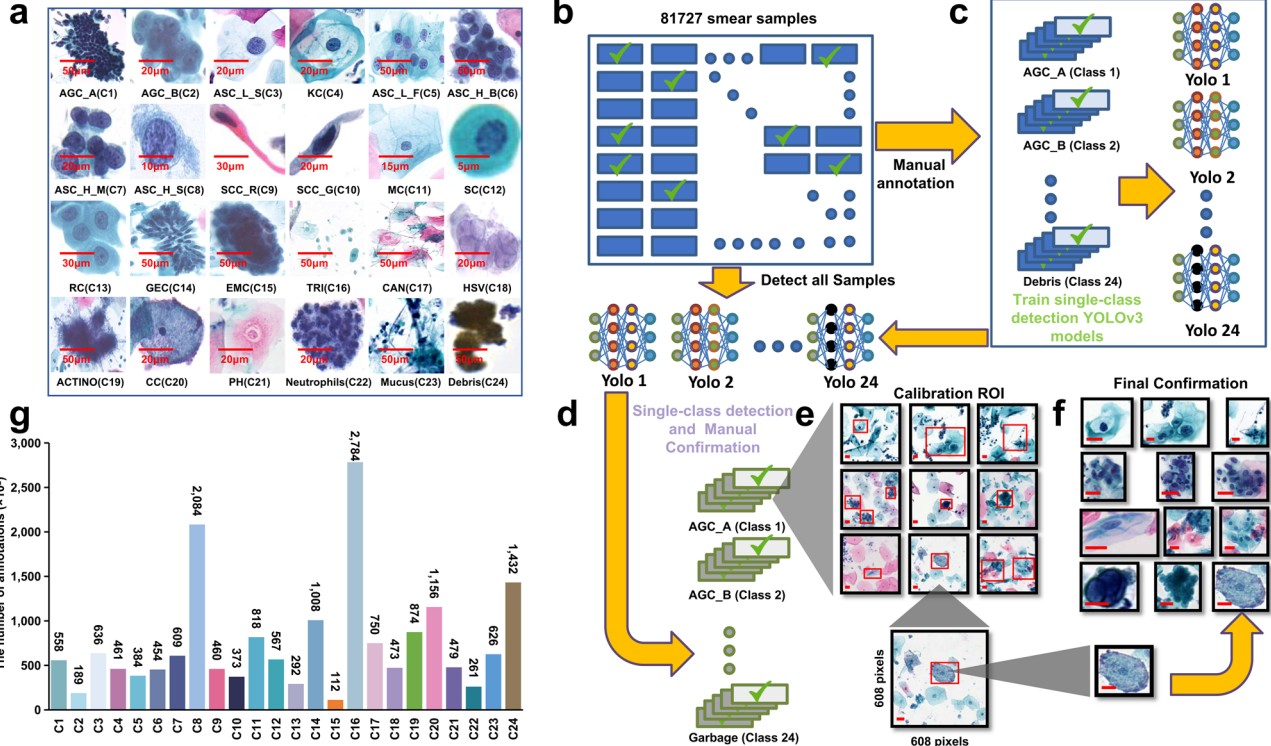

**Fig. 2 Obtaining large-scale annotation dataset for deep learning. a** Representative images of 24 new classifications according to the TBS standard (C1–C24 are the classification indexes during the learning). **b** Manually annotated 2000 images for each classification. **c** Training single-class YOLOv3 detection models for 24 classifications. **d** 81727 smears were detected by 24 single-class YOLOv3 detection models, and the detected targets were manually confirmed. **e** Cytologists calibrated the ROI of right targets using annotating software, and then the annotated dataset with the corrected ROIs were obtained. (Representative images from calibration process. Scale Bars: 50 μm). **f** Three cytopathologists confirmed the annotation data again. (Representative images from confirmation process. Scale Bars: 50 μm). **g** The annotation number of new 24 classifications (totally ~1.7 million annotated targets).

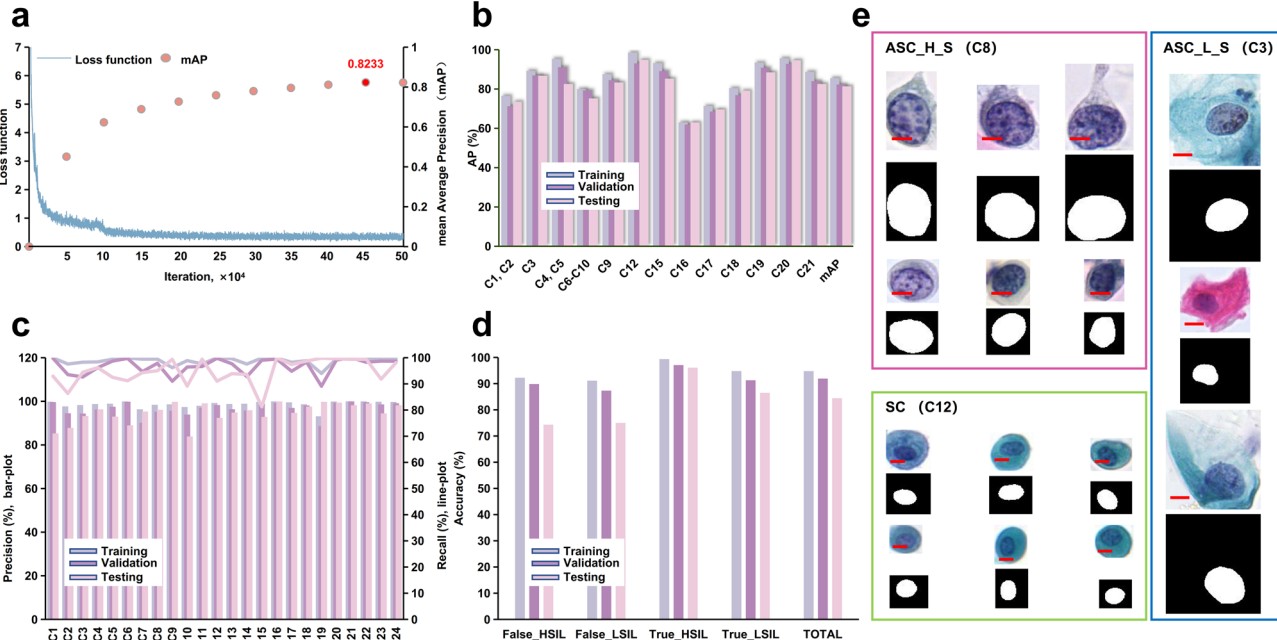

**Fig. 3 Training of the deep learning models and their performance. a** The loss function and mean Average Precision (mAP) of YOLOv3 model during the training. **b** The AP of 13 TBS classifications in YOLOv3 model. The corresponding classification indexes are shown in Fig. 2a and Table 1. **c** The precision and recall of 24 classifications in the Xception model. (Bar-plot: precision; Line-plot: recall). The corresponding names of each classification as show in Fig. 2a and Table 1. **d** The accuracy of False_HSIL, True_HSIL, False_LSIL, and True_LSIL in Patch model. **e** The representative images of ASC_L_S (blue box), ASC_H_S (red box), and SC (green box) in the nucleus segmentation model (true color image: original images, binary image: corresponding masks; Scale Bars: 10 μm).

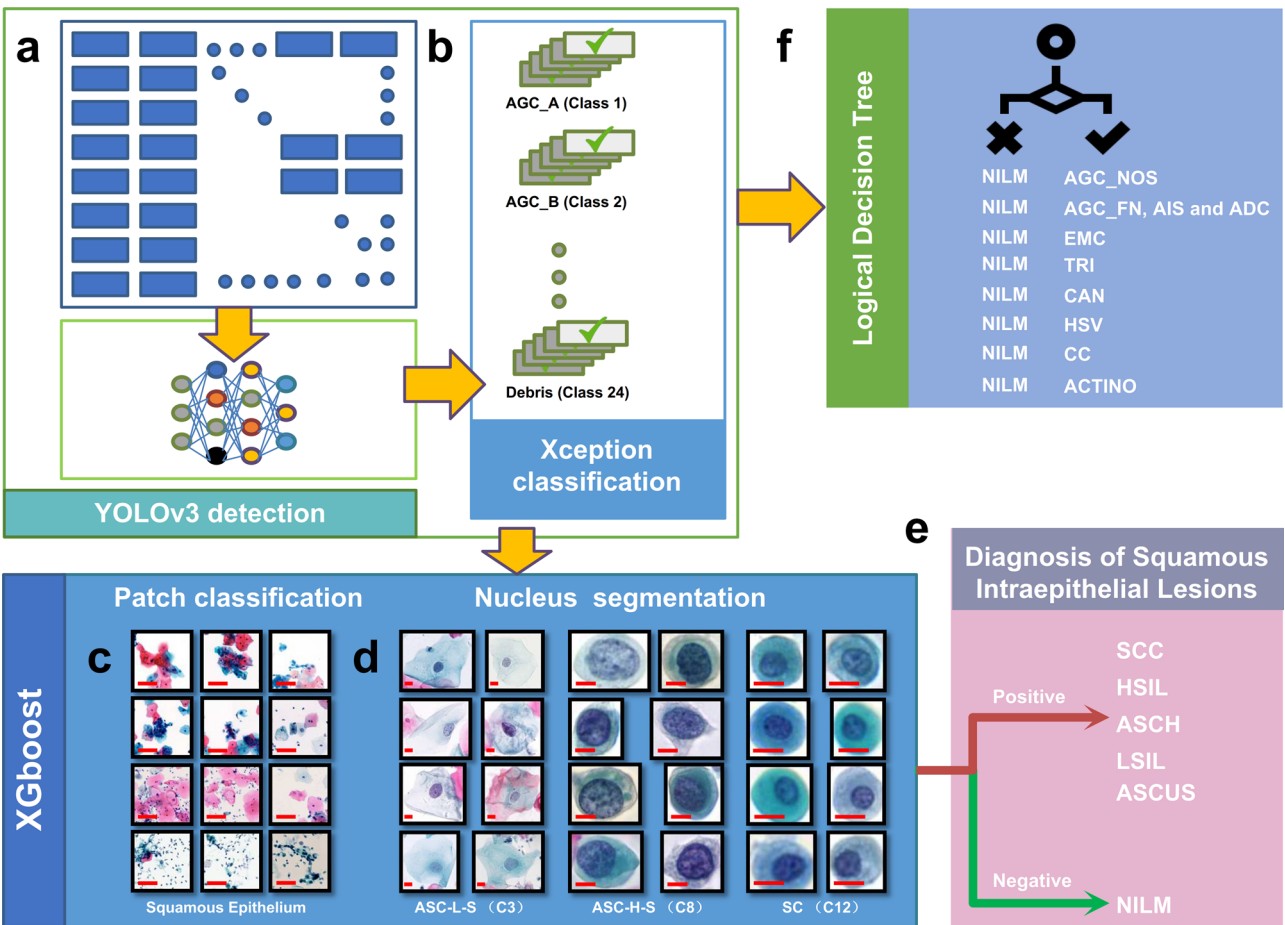

**Fig. 4 Diagnostic decision pipeline for TBS classification and diagnosis. a** All digital retrospective smears were detected by the Yolov3 model, and the classification and probability of the detection targets were extracted. **b** The Xception model predicted the targets detected by the YOLOv3 model, and then the information of classification and probability were extracted. **c** The Patch model was used to extract area classification and the probability of squamous epithelium targets. (Representative images in Patch model. Scale Bars: 50 μm). **d** Nuclear parameter extraction of ASC_L_S, ASC_H_S, and SC (Representative images in nucleus segmentation model. The length of Scale Bars in the images of ASC_L_S are 20 μm, while 10 μm in the images of ASC_H_S and SC). **e** Prediction for TBS classification of squamous intraepithelial lesions. **f** Logical Decision Tree was used to predict AGC, EMC, and Infectious lesions.

the TBS classifications are not uniform; for example, HSV and EMC are rare, while ASCUS, LSIL, and HSIL are common. Furthermore, the distribution of common intraepithelial lesions is not balanced as they are often associated with infectious lesions. Therefore, to achieve the final classification of smear according to the TBS diagnostic criteria, we delineated the common squamous intraepithelial lesions including ASCUS, LSIL, ASCH, HSIL, and SCC from other lesions and formulated the interpretation strategy separately.

We applied an XGBoost model[50] to distinguish between positive and negative squamous intraepithelial lesions as shown in Fig. 4a–d. We selected 121 features as shown in Supplementary Table 04, and they were the key to achieve efficient model recognition and differentiation. The selected features were input into the XGBoost model for further training. By testing the effectiveness of permutation and combination of DL models, we found that the AUC increased with the number of models and reached a maximum after all four models were combined, as shown in Supplementary Fig. 2a.

Then, we counted the number of splits of each feature in each tree, and after normalization, we obtained feature importance distribution histograms (Supplementary Fig. 2b). This result showed that the importance of the feature was related to the pathological characteristics of the lesion. For example, X-L13 and

X-SCC4 were the top two features in ratio of importance. The corresponding lesions expressing these features most highly, Koilocyte and keratinizing Squamous Cell Carcinoma, are two lesions with extremely obvious characteristics in cervical cytology. This feature importance study will be helpful to provide feedback to the cytopathologists and enhance their performance. The results of tenfold cross-validation of the models showed that the average sensitivity of the training samples was 83.96%, the specificity was 94.64% (Supplementary Table 05), and the average AUC obtained from tenfold cross-validation reached 96.73% (Supplementary Fig. 2c).

To increase the sensitivity of the AIATBS system for the application of screening, we fine-tuned the model, reduced specificity to a certain extent, and applied the ten models of the given tenfold cross-validation to predict the classification of the training samples. The threshold for determining a digital smear as a positive sample was the requirement that one or more of the ten models predicted that a given sample was positive. Finally, the sensitivity of the XGBoost model in the existing training set reached 100.00%, i.e., without missing any true positives, and the specificity was marginally reduced to 93.20% as shown in Supplementary Table 06.

With the previous XGBoost model, we could only predict the general cytopathological types such as squamous intraepithelial

lesions but not distinguish the classes required for the TBS diagnosis. To address this, we next excluded the negative classifications and classified positives into ASCUS, LSIL, ASCH, HSIL, and SCC. We then trained another XGBoost model to further classify squamous intraepithelial lesions into the subtypes required for TBS classification by using existing features as shown in Fig. 4e. We performed tenfold cross-validation (Supplementary Table 07) and refined the model with the highest accuracy of 80.43% as the TBS classification model for squamous intrae-pithelial lesions.

Infectious lesions, glandular intraepithelial lesions and endo-metrial cells have distinctive pathomorphological characteristics, so limited features are sufficient to achieve the required performance for classification. Therefore, we extracted the features from the YOLOv3 and Xception models (Fig. 4a, b) and used logical decision trees to make a final decision as shown in Fig. 4f and Supplementary Table 08. Because of the small number of samples containing glandular intraepithelial lesions combined with clinical treatment needs, we combined AGC_FN, AIS, and ADC into one class for prediction. We found that logical decision trees could clearly distinguish infectious diseases and severe intraepithelial lesions from negative samples as shown in Supplementary Table 09.

Overall, the results showed that by comprehensive application of DL and ML models, we could finally predict TBS diagnostic classification for WSIs of cervical liquid-based thin-layer smears.

**AI-based Digital Pathology Image Quality Control system is an important part of AIATBS**. Similar to the quality requirements of the TBS standard for smear samples, the quality of digitized image is crucial for AI model development. AI solutions may generalize to account for potential quality issues; however, such capability is not unlimited. The quality of smear samples depends on multiple links such as cervical exfoliated cell collection, sample preservation, smear production and dyeing. Obviously, for AI solutions, in addition to the above influencing factors, quality control in the smear digitization process is equally important.

To solve this problem, we designed and integrated an AI-based Digital Pathology Image Quality Control (DPIQC) system based on the XGboost model in our diagnostic platform as described in the "Methods" (Fig. 5). The average validation-set accuracy of our DPIQC system was 99.11%, and the detailed training and validation results are shown in Supplementary Tables 10, 11.

So far, after we successfully developed the DPIQC system, our AISTBS system includes the following for AI models: (I) YOLOv3 detection model, (II) Xception classification model, (III) Patch area classification model, (IV) Nucleus Segmentation model, (V) XGBoost model and Logical Decision Trees, and (VI) DPIQC system. Different models play their role and were integrated into the AIATBS system to achieve accurate TBS classification and quality control of digital smears. The strategic integration of multiple learning models is described in the Methods, and the flowchart of our AIATBS system platform is shown in Supplementary Fig. 3.

**AIATBS system permits accurate and rapid TBS classification**. After the training and optimization of our AIATBS system, we were ready to conduct a clinical trial including 11 medical institutions in China. As image quality control was a key step before clinical annotation and training, ~9.6% of samples were unqualified and rejected by using the DPIQC system. It is worth noting that 89.11% of these unsatisfactory smears excluded by the DPIQC system were also excluded by the senior cytologists (Supplementary Table 12). We then collected a total of 34,403 prospective cervical liquid-based cytology smears accepted by the DPIQC system for verification. The smear preparation methods again included natural, membrane and

centrifugal sedimentation, and the staining schemes included both EA-36 and EA-50 (Fig. 1b).

Based on the final TBS classification results as shown in Fig. 1d (diagnostic confirmation standard was given in the Methods), we analyzed the sensitivity and specificity of the AIATBS system in samples of different sample preparation, staining schemes, and digital scanning. The statistical analysis indicated that the AIATBS system had good sensitivity and specificity in smear sample sets of different sample preparations (sensitivity: $p = 0.4108$, specificity: $p = 0.5773$) (Fig. 6a), different staining schemes (sensitivity: $p = 0.3268$, specificity: $p = 0.2442$) (Fig. 6b), and two different scanners (sensitivity: $p = 0.0148$, specificity: $p = 0.0839$) (Fig. 6c). These results suggested the sensitivity and specificity of the AIATBS system were related to the type of scanners.

To evaluate the performance of the system, we adjusted the statistical classification according to the clinical treatment principles of cervical precancerous lesions. We combined LSIL, ASCH, HSIL, and SCC in the classification of squamous intraepithelial lesions and analyzed the glandular intraepithelial lesions according to the classification of the AIATBS system. Our data showed that the sensitivity of the model for detecting intraepithelial lesions and other lesions (including infectious lesions and EMC) was 92.00% and 83.00%, respectively, and the overall specificity was 82.14% (Supplementary Table 13), where the false negative and false positive samples were mainly in ASCUS. Meanwhile, we found that the sensitivity and overall specificity of the AIATBS system in medical institutions that previously provided retrospective samples were slightly higher than those in medical institutions that did not provide retro-spective samples, however without statistically significant differ-ence (sensitivity of intraepithelial lesions: $p = 0.2751$, sensitivity of other lesions: $p = 0.2616$, overall specificity: $p = 0.2398$) (Supplementary Table 14). The accuracy of the XGBoost model in predicting the corresponding TBS classification of squamous intraepithelial lesions was 74.15% (Supplementary Table 15).

The average computational time for diagnosis of natural, membrane and centrifugal sedimentation was 66.27 s, 171.8 s, and 85.30 s, respectively ($p < 0.001$) (Supplementary Table 16). The natural sediment has the shortest diagnosis time. For the devices, the average time for diagnosis of each sample scanned by scanners 1 and 2 were 89.30 s and 107.20 s, respectively, ($p < 0.001$). If we fix the scanner type (Scanner 1) and smear preparation (natural sedimentation), the average diagnosis time for each sample stained by EA-36 and EA-50 protocol were 67.25 s and 68.69 s, respectively, ($p = 0.0657$), as shown in (Supplementary Table 16). The possible factors which may impact the computational time of AIATBS are the sediment method and the scanner, but not the staining solution.

We conducted statistics on the manual diagnosis of senior Cytologists in six medical institutions (indicated by "Medical institutions a–f" in Fig. 6d), and compared them with the results of the AIATBS system. The results showed that the sensitivity of senior Cytologists to manually detect intraepithelial lesions and other lesions (including infectious lesions and endometrial cells) was 86.12% and 76.33%, respectively, which were lower than the 90.75% and 84.23% of the AIATBS system ($p = 0.0435$, $p = 0.0254$), and their specificity was 90.18%, which was higher than the 81.93% of the AIATBS system ($p = 0.0131$) as shown in Fig. 6d. The senior Cytologists' accuracy (true positive rate) of TBS classification for squamous intraepithelial lesions was 80.16% (Supplementary Table 17), while the accuracy of the AIATBS system was 75.24% ($p = 0.0984$) (Supplementary Table 18). The main reason for the low accuracy of the AIATBS system is that HSIL and ASCH are not easy to distinguish. SCC is easily misjudged as HSIL, but this variation has no effect on the subsequent clinical treatment and is acceptable in clinical

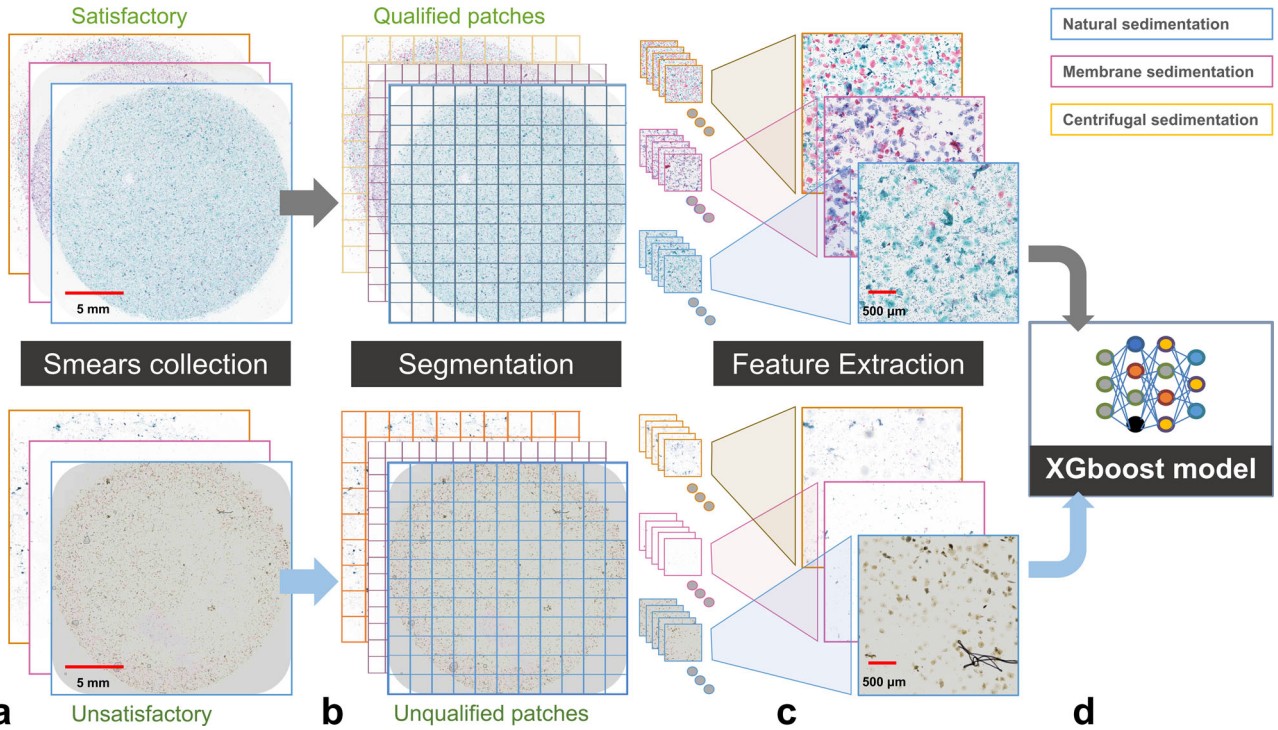

**Fig. 5 Development of the efficient AI-based Digital Pathology Image Quality Control (DPIQC) system. a** Screening satisfactory and unsatisfactory digital smears produced by three different sedimentation methods. **b** Division of the whole digital smears into patches with a size of 6000 × 6000 pixels. **c** Feature extraction of the patches. **d** Trained XGBoost model for predicting whether digital smear was qualified. The representative images in (**a**–**c**) display the construction process of the DPIQC system.

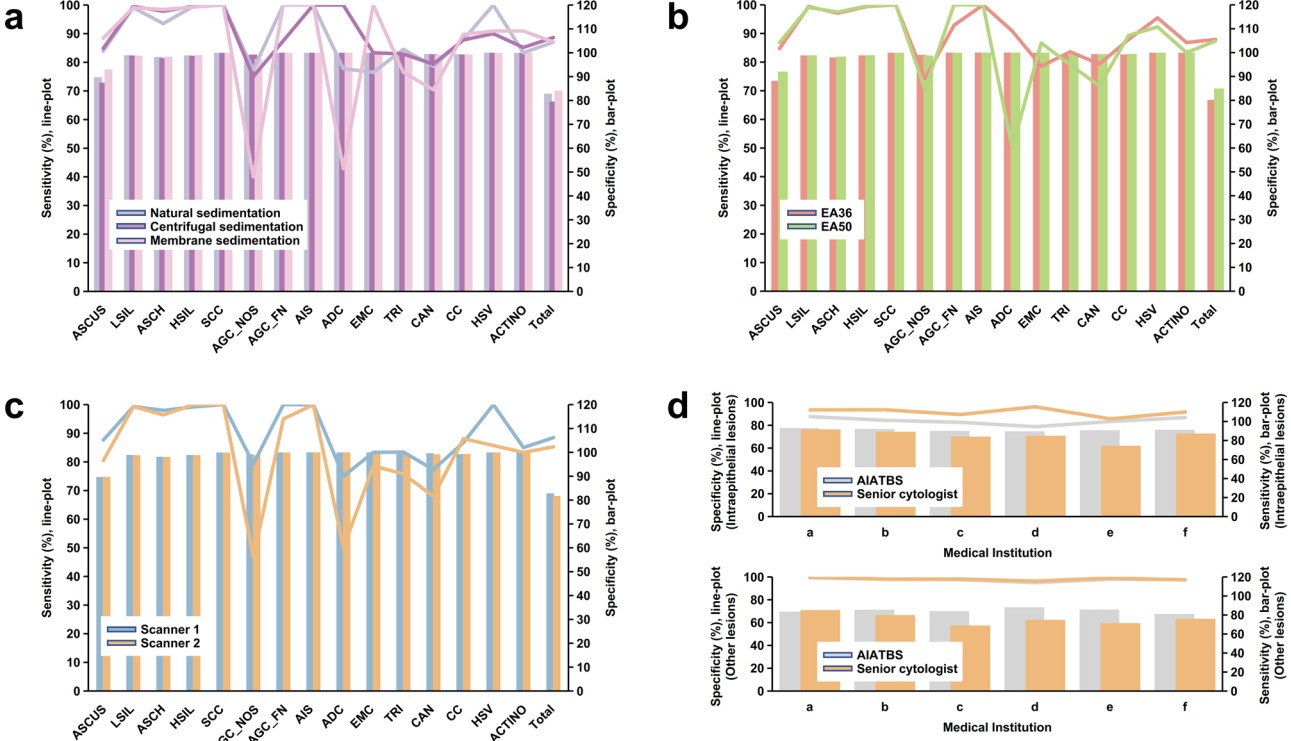

**Fig. 6 AIATBS system achieved good generalization and showed higher sensitivity than senior cytologists. a** The sensitivity and specificity of AIATBS system in predicting three different preparation smears. (Bar-plot: specificity; Line-plot: sensitivity). **b** The sensitivity and specificity of AIATBS system in predicting two different Papanicolaou stain smears. (Bar-plot: specificity; Line-plot: sensitivity). **c** The sensitivity and specificity of AIATBS system in predicting smears which digitized by two scanners. (Bar-plot: specificity; Line-plot: sensitivity). **d** Comparison of sensitivity and specificity between senior cytologist and AIATBS system in diagnosing cervical liquid-based thin-layer cell smears. (Bar-plot: sensitivity; Line-plot: specificity).

**Table 2 The sensitivity of AIATBS system in cervical histological biopsy diagnosis.**

| Medical Institution | Biopsies number | Positive sample number | AIATBS missed sample number | AIATBS sensitivity |
|---|---|---|---|---|
| I | 180 | 145 | 8 | 0.944827586 |
| II | 382 | 298 | 17 | 0.94295302 |
| III | 418 | 394 | 19 | 0.95177665 |
| Total | 980 | 837 | 44 | 0.947431302 |

practice. In our prospective database, we obtained 980 cervical liquid-based cytology smears with corresponding pathological biopsy diagnosis results (which serve as ground truth) from three large medical institutions and compared these results with the AIATBS-based cytology diagnoses of smear samples. Through calculation, we found that the sensitivity of AIATBS system for cervical intraepithelial lesions was 94.74% as in Table 2.

Overall, the sensitivity of our system was higher than that of senior Cytologists while still maintaining high specificity. These results shows that the AIATBS system is able to improve the screening diagnostic quality of cervical liquid-based thin-layer cell smears in clinical practice and reduce the Cytologists' workload.

## Discussion

In the past few decades, the mortality rate of CC in many countries has been on an overall downward trend[51,52], which is largely due to development of CC examination methods including the Pap smear, colposcopy, and human papilloma virus (HPV) screening. The Pap smear[53] is the most frequently employed method, but it yields a high false negative rate for CC screening. The liquid-based Pap tests have been suggested as alternative cervical screening methods[54]. Various preparations of cervical liquid-based thin-layer cell smears have greatly reduced the false negative rate of CC screening and have laid a solid foundation for early diagnosis and treatment. In China, there is an extreme shortage of cytologists, and the workload of CC screening is huge. Cytologists need to spend a long time looking for various lesions from thousands of cells in each smear, a process prone to missed diagnoses. The AIATBS system is developed to reduce their workload, improve the accuracy of diagnosis and solve these unmet needs.

Cervical cytology is the most researched direction in the field of pathology, involving many researches such as image recognition, classification, and cell nucleus segmentation[22]. Our system integrates multiple DL models, other ML models and a logical decision tree to achieve digital cervical liquid-based smear detection, classification and nucleus segmentation tasks to reasonably formulate a TBS diagnosis decision based on WSI. At present, the intelligent screening systems for regional detection of cervical liquid-based cytology smears[31,33] implement risk stratification management of cervical lesions, which makes it possible to control the diagnostic quality. However, these solutions mainly focus on cervical intraepithelial lesions and cannot classify and grade the lesions, and they ignore infectious lesions and thus cannot perform auxiliary diagnostics strictly according to the TBS standard in clinical screening. We compiled a large dataset, collecting >81,000 retrospective cervical liquid-based thin-layer smears prepared according to a range of standards and staining protocols from multiple centers and annotated the smear strictly according to the requirements of the TBS criteria. It is one of the largest and most comprehensive studies of AI-based cervical liquid-based cytology smear diagnosis. We have developed the AIATBS system and performed extensive verification experiments to validate its performance in prospective clinical decision-making. The effects of these results are extensive: (I) we classified the lesions in more detail according to their morphology, significantly improved the accuracy of detection and

classification and provided a nucleus segmentation model for accurate quantification of cell nuclei, (II) we used an efficient semi-supervised tagging model, which not only ensures the accuracy of data but also rapidly expands the amount of data (III) by integrating the multiple DL models in AIATBS, it extracts different levels of features to complement each other, effectively quantifies the classification objectives, and provides accurate parameters for the construction of a strongly generalized model to predict the TBS classification and diagnosis of the whole digital smear (IV) the ML and logical decision tree, based on the diagnostic experience of cytopathologists, can truly achieve TBS classifications of cervical liquid-based cytology smears, and (V) We also developed the AI-based DPIQC system to perform good quality control in the preparation, staining and scanning stages of cervical liquid-based cytology smears. The above five strategies allowed us to develop and train our AIATBS for cervical liquid-based thin-layer cytology, which can be extended to pathological practice for prospective data and integrated into the clinical workflow of CC screening.

In the clinical prospective validation, AITBS system showed the characteristics of high speed (<180 s/smear), high sensitivity (intraepithelial lesions ≥92.00%, other lesions å 83.00%) and high specificity (82.14%), while also showing excellent generalization performance for all kinds of smears and staining protocols. In particular, it is worth noting that AITBS shows a higher sensitivity than senior cytologists in prospective studies. The follow-up results of cervical histological biopsy also showed 94.74% sensitivity, which lays a solid foundation for the clinical application of the system.

However, we also found some problems that are worth noting. First of all, most of the false negative and false positive cases were ASCUS (Supplementary Table 13), suggesting that similar to a cytopathologist's diagnosis, the model also has lower accuracy for this challenging category (<90%), so it is necessary to accumulate more training data and correct those errors through HPV detection and histological biopsy. Secondly, the system had some difficulty detecting cervical glandular epithelial atypical hyperplasia and adenocarcinoma. The reasons may be the lack of training data for glandular epithelial lesions, poor smear preparation, staining and scanning. The solution to these problems lies in the accumulation of more training data and strict quality control of smear preparation, staining and scanning.

Finally, the developed AIATBS system is strictly based on the requirements of the TBS standard to pre-test the whole smear and shows strong predictive capability. It is expected to effectively assist most of the screening work of senior cytologists, thus greatly reducing the work burden of senior cytologists and cytotechnologists, with broad application prospects for assistive diagnosis.

## Materials and methods

**Ethical approval.** The retrospective and prospective studies were approved by the Medical Ethics Committee of Nanfang Hospital of Southern Medical University (Ref. No. NFEC-2019-241). The informed consents were waived by the Medical Ethics Committee since the samples were irreversible anonymised.

**Smear sample digitization and establishment of TBS classification diagnosis.** All smears were scanned with 0.25 µm/pixel resolution using two different scanners (Scanner 1: linear scanning and single layer; Scanner 2: area array camera scanning and single layer). For the retrospective study, the positive (including intraepithelial lesions, infectious lesions, and EMC) smears were reviewed by three

cytopathologists who diagnosed according to the TBS standard. For intraepithelial lesion smear, they first referred to the histological diagnosis of the cervical biopsy corresponding to the smear. For smears that had no histological results or false negative histological results which might be caused by irregular cervical biopsies, the unanimous diagnosis of the three cytopathologists prevailed. For smears that could not reach a unanimous diagnosis, the positive diagnosis with the highest lesion grade prevailed. To avoid the more challenging case of false negatives from subtle pathologies, the negative (NILM) smears were reviewed by senior cytologists. Negative smears were evenly distributed to 20 senior cytologists for reexamination, and three cytopathologists randomly checked 10% to ensure that there were no positive cases in negative smears.

**Multicenter retrospective smears annotation.** Based on the TBS diagnostic criteria, the WSI of the digital smears with 24 different classifications (as shown in Fig. 2a) were manually annotated by 20 senior Cytologists using the annotation software known as Automated Slide Analysis Platform (ASAP)1.8 (Supplementary Fig. 4a), an open-source platform for visualizing, annotating, and automatically analyzing whole-slide histopathology images.

The annotation was carried out along the boundary of each lesion area, with the width and height of the image on either side having no more than 608 pixels. An annotation file was then generated and recorded. It contained the necessary structured information for the following AI learning, such as the category information and a rectangular Region of Interest (ROI) image, which was a rectangular region defined by the coordinates in the scanned images. If the width or height of any image exceeded 608 pixels, the senior cytologists decomposed the image in to several small images according to the morphological structure of the lesion area such that the width or height of the decomposed images were within the given range (Supplementary Fig. 4b).

In order to accelerate the data annotation, a pretrained single-class YOLOv3 model, known for its accuracy and efficiency, was created based on the above manually annotated data as shown in Fig. 2b, c. The pretrained model detected the lesion classifications and generated rectangular ROIs for the following semiautomated annotation and AI learning. Afterward, senior cytologists further assessed the detected images and eliminated any non-targeted ROIs (Fig. 2d).

Finally, patch files comprising 608 × 608 pixels were generated from the smears for each ROI obtained from one-class YOLOv3 models. Used labeling software to calibrate the incorrect ROI of each image (Supplementary Fig. 4c and Fig. 2e), the cropped patch images for each ROI were annotated into different defined categories containing relevant information (Supplementary Fig. 4d). The cropped patch files were saved, and the annotation dataset was reviewed and confirmed again by cytopathologists (Fig. 2f).

**Deep learning models for target detection, fine-grained classification, patch classification, and nucleus segmentation.** After annotation and verification by the cytopathologists, we combined our training dataset with SC and PH except GEC, MC, RC, Neutrophils, Mucus and Debris classification, and merged similar classifications to one class in pathomorphology such as C1 and C2, C4 and C5, and C6-C10. Next, 1216 × 1216 pixel ROIs were created, each large enough to include neighborhood information, according to the coordinate position and then uniformly resized to 608 × 608 pixels. The target detection training was organized based on the Darknet53 framework, and a YOLOv3 detection model were obtained. The training process iterated for 450,000 epochs until the mAP of the test set no longer improved (Fig. 3a).

The annotated images of 24 annotated classifications were normalized to 299 × 299 pixels by using bilinear interpolation. Then Xception classification model was applied for the classification training. A total of 18 epochs were iterated until the accuracy of the verification set could no longer improve. The images from each different classification detected by YOLOv3 were input into the Xception model to extract refined features for classification.

ASC_L_S, KC and ASC_L_F in the annotated data with classification of squamous intraepithelial lesions were merged as True_LSIL. ASC_H_B, ASC_H_M, ASC_H_S, and SCC_G were merged as True_HSIL. The corresponding False LSIL and False HSIL classification data were composed of false targets corresponding to the Xception classification detected by the YOLOv3 model from the negative smears. We used this data to train the Patch classification model. The training data was organized using ROIs of 608 × 608 pixels with information around the targets for the feature selection. The ROIs were convoluted by the two-dimensional Gaussian kernel using DenseNet-50. The output were the selected features designed for the positive and negative decision, such as True_HSIL, False_HSIL, True_LSIL, and False_LSIL. We applied iterative training based on the cross-entropy loss function until the loss of the testing set no longer decreased. The model with the highest accuracy in the tenfold cross-validation training was selected as the final prediction model. When the model predicted the target, we used the Sigmoid function $[\sigma(z_i) = \frac{1}{1+e^{-z_i}}]$ to output the probability value of the corresponding classification, thereby mapping the value of the classification probability to [0–1].

The training data for a nucleus segmentation model was composed of images annotated and classified as SC, ASC_L_S and ASC_H_S. The quantity ratio of each type of image was roughly 1:1:1. 80% of the dataset was used for training and 20% for validation. The augmentation data was performed by rotating, exchanging color channels, adjusting brightness, contrast and increasing the noise, etc. The nucleus segmentation model was trained using a U-Net algorithm. Dilated convolution was used for the convolution kernel. Deconvolution was used for up sampling, and the loss function was Dice loss + cross-entropy loss[55]. After 50 epochs of training iteration for the validation set, mIOU no longer improved, and the training was terminated. The highest mIOU model from fivefold cross-validation was selected for the final segmentation. After the cell nuclei of targets were segmented by the model, the target images were converted to grayscale, and the gray value of the cell nuclei were calculated (Supplementary Fig. 1b).

In order to build and train the above DL models, all the code was written in Python (3.6). We used Open Slide(0.4.0), a C library that provided a simple interface, to read whole-slide images. TensorFlow (1.13.1) and Mxnet (1.5.1) were used to train DL models and do network inferrence.

**Machine learning model training and logic decision tree.** According to the optimized features and output of the above DL models, the feature selection was mainly based on the cytopathologist's diagnostic experience according to the TBS standard, i.e., features known to be defining based on clinical experience, while other features were selected according to simple logistic regression, i.e., algorithm features. The 121 features obtained from the YOLOv3 detection model (Fig. 4a), Xception classification model (Fig. 4b), Patch classification model (Fig. 4c), and nucleus segmentation model (Fig. 4d) were input into an XGBoost model for diagnostic model training. Prediction results of positive and negative squamous intraepithelial lesions were reached. Next, the positive results were further classified by a simple XGBoost model for squamous intraepithelial lesions TBS classification as shown in Fig. 4e.

The features of infectious lesions, endometrial cells and glandular intraepithelial lesions were collected from the YOLOv3 and Xception models (Fig. 4a, b). First, we used a shallow decision tree to preliminarily define the above classifications to obtain thresholds with high sensitivity and specificity, and then according to the distribution of feature values of each classification in the training set, the cytopathologist supplementally delineated some thresholds to predict above classifications based on the TBS criteria, so that the logic decision tree strategy of the TBS classification diagnosis could reach the highest sensitivity and specificity as much as possible (Fig. 4f and Supplementary Table 08).

The final TBS classification diagnosis of cervical liquid-based thin-layer cell smear was obtained by combining the XGBoost model with the logical decision tree, i.e., final diagnostic decisions, as shown in Supplementary Fig. 3.

**Strategic integration of multiple learning models in the AIATBS system.** In our research, the most important task in the construction of multiple DL models is to provide accurate parameters for the ML and logical decision tree diagnosis model to predict the TBS classification of the whole smear. In the training data, this purpose is achieved by extracting complementary training of different levels of features, monitoring training, and verifying the loss and accuracy of the data. The YOLOv3 target detection model is used to detect lesion targets, and the annotated categories with relatively consistent pathological morphology are merged, so that YOLOv3 can learn the commonness of different types of morphology and improve the efficiency of detection. The classification probability obtained by YOLOv3 was affected by the target background, target size and noise around the target. We used a further accurate classification of the detected target to extract high-precision target features. Next, we input the detected targets into Xception[47], which is more efficient than Inception V3, for fine feature extraction, and classification. Since there are still a large number of false targets detected by YOLOv3 model after the Xception model classification, we used the Patch model to further identify squamous intraepithelial lesions in CC screening. The Patch recognition module selects the Densenet model with deeper layers, which can effectively express the deeper semantic features of the image. This approach not only retained more morphological and dimensional information of cells[48,56], but also introduced a Gaussian distribution highlighting the target cell area from its surrounding area, enabling the network to focus on a small-dimensional target region and extract features effectively. In addition, nuclear enlargement is the most important marker of cervical intraepithelial lesions. Indeed, DNA ploidy technology used to screen cervical precancerous lesions and make an auxiliary diagnosis is based on this cellular feature[57]. In order to obtain the parameters of the size of the squamous cell nucleus, we use the U-net network[58,59] which is widely used in medical image segmentation, to segment the target nucleus and calculate the gray value. The training results show that the quantitative parameters of the segmented nuclei constitute an important part of the parameters of the XGboost model for classifying squamous intraepithelial lesions. AIATBS integrates several DL models and still can quickly achieve target detection and classification mainly because YOLOv3 directly predicts the target location and classification information, accelerating detection speed over that of Faster R-CNN and RetinaNet[43–45]. Similarly, Xception can also accelerate the prediction speed through batch prediction.

One of the main challenges of DL is over-fitting, because over-fitting models cannot be extended to unseen data. We used the XGBoost model (squamous intraepithelial lesions) and logical decision tree (infectious lesions, glandular epithelial lesions and endometrial cells) to integrate the parameters extracted from multiple DL models to predict the TBS classification of cervical liquid-based thin-layer cell smears so as to simulate the process of TBS reports made by

cytopathologists after observing various lesions. The XGBoost model employed a random forest for reference, took into account the probable situation of sparse training data, and specified the default direction of branches for missing values or specified values, which can be lower than fitting[50]. It could solve the problem of poor generalization caused by the inconsistent characteristics of various types of lesions. The training results showed that the model could accurately predict the squamous intraepithelial lesions and make a relatively correct TBS classification for diagnosis. For certain TBS classifications that are represented by a small number of cases and typical morphology, clinical judgment by cytopathologists according to diagnostic experience was needed to avoid false negatives. Of course, the best way to reduce over-fitting is to get more training data, and we continue to collect more clinical data to optimize the system.

**Digital pathology image quality control (DPIQC) system**. We scanned the smears (including preparation of natural sedimentation, membrane sedimentation and centrifugal sedimentation) with 0.25 μm/pixel resolution digital pathology WSI scanners (Fig. 5a) and segmented the digitized images into $6000 \times 6000$ pixel images (Fig. 5b). The training data for DPIQC system, which contains both satisfying/unsatisfying images, was therefore carefully selected and confirmed by three cytopathologists together. Then, we used the Laplacian operator to convolve the block and calculate its gradient to evaluate the images focus. The images were transformed into hue, intensity and saturation (HIS) space, features of the images were extracted from various dimensions, and feature histograms were calculated to generate input for contrast evaluation. In order to evaluate whether the given large image contained enough cell quantity for diagnosis, we implemented a cell counting module to roughly evaluate the number of cells in the entire large image. The specific method was to use the Otsu threshold segmentation method to separate cell and non-cell regions and then calculate the ratio of the cell area to the entire image as an evaluation indicator. We extracted the above features from qualified and unqualified images (Fig. 5c) and input the features into the XGBoost model to build the DPIQC system (Fig. 5d).

**Multicenter prospective study for AIATBS system**. Senior cytologists assessed the smears according to the clinical diagnosis protocol, and the automatic analysis was carried out using our developed online AIATBS or local server (server hardware configuration given in Supplementary Table 19). The experiments were double-blind for both the senior cytologists and the AIATBS system. The diagnostic results obtained manually by senior cytologists, the classification results of our AIATBS and the time taken were recorded. The performance, including diagnostic efficiency and effectiveness of AIATBS and senior cytologists, was evaluated. The protocol of determining the final diagnosis was the same as the retrospective study as described above. As the classification of squamous intraepithelial lesions and intraglandular lesions is prone to error in cytopathological diagnosis, but this error does not affect clinical treatment; as long as the AIATBS system predicted squamous epithelium lesions or glandular epithelium lesions as intraepithelial lesions in the TBS classification, we did not consider the error of this classification in sensitivity analysis. In addition, ~1.2% of smears with two or more lesions were classified as a single lesion according to the priority of the clinical treatment required for the lesion.

**Reporting summary**. Further information on research design is available in the Nature Research Reporting Summary linked to this article.

## Data availability

In this work, we presented a hybrid AI-assistive diagnostic model for TBS classification of cervical liquid-based thin-layer cell smears. This paper was produced using no publicly available image data as it is constrained by personal information protection, patient privacy regulation, and medical institutional data regulatory policies, etc. The size of our research data is also too huge to be properly stored in public repositories. However, the authors have made every effort to make the available of these resources publicly available such as the source code, software methods and the supporting information to reproduce technical pipeline, analyses, and results. All data supporting the finding of this work are available unconditionally for accredited scientific researchers for the purpose of reproducing the results and/or further academic and AI related research activities from the primary corresponding author dyqgz@126.com upon request within 10 working days.

## Code availability

The source code for training the models mentioned in this work is available at https://gigantum.com/louguei/ncomms, or obtained by sending a request to the primary corresponding author (Prof. Y.D., dyqgz@126.com).

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

## Acknowledgements

The authors would like to thank Longshan Zhang, Xixi Wu, Chongchong Zhang, Yu Ye, Chuan Wang, Qifeng Wen, Jinfeng Zhan, Chuqi Zheng, Tiejun Guo, Guifang Ding, and Han Chu for data processing and helpful suggestions. This work was supported by the fund program: Development and application demonstration of AI-assisted pathological diagnosis system for cervical liquid-based cytology (Guangzhou R & D plan of key areas, Key Areas Research and Development Program of Guangzhou, 202007040001). This work is jointly supported by BII and IMCB, BMRC, A*STAR research funding.

## Author contributions

X.Z., W.Liao, Y.D., L.Liang, Y.S., J.S., and W.Y. designed the experiments; W.Zhang, X.Li, W.X., Y.Liu, W.Li, Y.Z., N.T., A.H., H.G., and Z.C. collected the smears and contributed to their final diagnosis; X.Z., W.Zhang, X.Li, Y.D., and L.Liang helped identify intrac cases; X.Z., F.W., Q.L., S.L., M.L., Y.Li, W.Zhong, and X.Liang manually labeled the dataset; X.Z., W.Zhang, X.Li, J.X., Y.Z., N.T., M.L., Y.Li, W.Zhong, and X.Liang reviewed the dataset detected by YOLOv3 model and calibrated the ROI of images; B.S., L.P., Y.S., J.S., L.R., J.L., and Z.Y. wrote the code to achieve different tasks; X.Z., K.O., L.Li, H.H., Y.S., and W.Y. contributed to the analysis of the data; D.Y., Y.D., L.Liang, and W.Y. conceived and directed the Project; X.Z., W.Liao, L.Liang, Q.L., and W.Y. wrote the paper with the assistance and feedback of all the other co-authors.

## Competing interests

The authors declare no competing interests.

## Consent for publication

All authors of this work agreed to publish with Nature Communications once accepted.
