## [Peer Review File · Nature Communications]

Reviewers' Comments:

Reviewer #1:

Remarks to the Author:

AI is a promising technology that could improve the diagnosis workflow in real clinical practice. This paper presented a computerized method to automatically process and classify digitalized TCTs based on the Bethesda system. My major critiques are:

1. In the 5-fold cross validation, the XGBoost model did not perform well in distinguishing ASCH, ASCUS, HSIL, LSIL, and SCC. For instance, LSIL had about 65% of accuracy. HSIL obtained the accuracy from about 66% to 73%. These values of accuracy also had a large variation. This might suggest that the model could not work properly with a low stability. The model robustness needs to be evaluated further.
2. Why was the 5-fold across validation used? In general, 10-fold across validation is commonly adopted. By the other hand, the collected dataset is large enough to split into training and test dataset. The model could be well tested on an independent test dataset.
3. How large the variations between the data acquired from 11 hospitals? Did the variations degrade the model's performance?
4. Since multi-center data cohorts are used in this paper, it is a better way to build a model, which is trained on one cohort and tested on another one. Therefore, the model is well validated on independent dataset.
5. The AI-based classification could be compared with the human cytopathologists in terms of classification accuracy, sensitivity, and specificity.
6. After nuclear segmentation from the U-Net model, a total of 121 features were extracted from the segmented nuclei. Did these features match the TBS standard, in which morphological features are usually reviewed by the pathologists? What was the feature selection method used in the presented pipeline? Did this feature selection method work well to identify the important features? Could these selected features be useful to assist the pathologists during the diagnosis procedure?
7. The current quantitative results could not sufficiently support the authors' conclusion without more experiments and systematic analysis.

Reviewer #2:

Remarks to the Author:

This manuscript describes a system developed to classify liquid based cell smears for early detection of cervical cancer according to the Bethesda system. The major contribution from this study is that they address the whole screening problem from handling specimens prepared at different labs through automatically finding and removing poorly prepared specimens, finding relevant cell clusters and classifying those based on general appearance through trained neural networks but also through segmentation of nuclei and extraction of features that can be used for refining the classification finally confirming the developed system on a completely separate dataset. Very many papers are published addressing some of the problems e.g. segmenting or classifying cells but very few studies addresses the whole automated screening problem. Also most studies deal with a few hundred samples at most. A really strong feature is that this study is based on a massive number of cases, >81 000 for the training and >34 000 in the test set. The results show performance on the same level as human cytopathologists for the same material. The study is thus an important contribution to the state of the art in the field. It does however suffer from a number of issues that need to be addressed before publication.

My most serious criticism of the paper is that the authors seem to have a rather vague idea about the history and state of the art in their field. This is particularly clear from the literature review in the introduction.

At line 138 it is correctly stated: Cervical cytology is an early application of computer-aided diagnosis and has applications to a wide range of cancer pathologies. But only one reference (reference 20) to one particular study in the field is given. This does in no way acknowledge the

fact that image analysis assisted cervical cancer screening is the most studied of all applications in cytology and histopathology, probably in medicine as a whole. The first studies were done in a major US project in the 1950-ies see "W. E. Tolles and R. C. Bostrom, "Automatic screening of cytological smears for cancer: the instrumentation", Annals of the New York Academy of Sciences, vol. 63, 1956." Since then many thousand papers have been published. There were commercial systems for the screening available from the 1990-ies. One of many review articles describing the history of the field is "Screening for Cervical Cancer Using Automated Analysis of PAP-Smears." Bengtsson and Malm, Computational and Mathematical Methods in Medicine", 2014.

This lack of meaningful referencing of the literature in the field is also apparent in the description of the more recent work where AI technology has been applied to this problem. The references describing the state of the art in deep learning applied to pathology or cytology diagnosis seem to be rather randomly chosen. Not particularly focused on the kind of applications the manuscript is about.

Reference 12 published in 2009 does not as stated describe "AI solutions were also used in cytological screening, quantitative morphological analysis, histopathological diagnosis and prognosis prediction" since it was published in 2009, before the modern boom in AI deep learning technology started. There were work on neural networks applied to this problem far before the current boom see e.g. L. J. Mango, "Computer-assisted cervical cancer screening using neural networks," Cancer Letters, vol. 77, 1994, but they were not discussed in reference 12 and due to the limited computational power available at that time they were not as successful as more traditional image analysis approaches.

A more recent review that does address also AI based approaches is: Image analysis and machine learning in digital pathology: Challenges and opportunities. Med Image Anal. 2016. An even more recent and more comprehensive review is: Deep learning in image cytometry: a review. A Gupta et.al. Cytometry Part A, 2019, it includes papers on cervical cytology screening. Another recent and highly relevant review is: Artificial intelligence in cytopathology: a review of the literature and overview of commercial landscape. Landau and Pantananowitz. Journal of the American Society of Cytopathology, Vol 8, 2019. Even after the publication of those reviews a number of highly relevant papers have been published in the very active field of deep learning applied to cervical cytology screening.

There are also a number of other strange statements in the introductory literature review where the titles do not seem to fit the topic of their citations. For instance reference 13 is said to deal with lung cancer although the actual topic is colorectal cancer. So the whole description of the field and the literature review need to be significantly updated.

At Line 84 it is stated: This global success is mainly attributed to the establishment of cervical cancer screening projects based on the Thinprep Cytologic Test (TCT) technology. TCT has laid a solid foundation for early diagnosis followed clinical treatment.

Similarly at line 404 you state: Since the 1970s, the mortality rate caused by cervical cancer in many countries has been on an overall downward trend, which is largely due to the development of cervical liquid-based thin-layer cell smear technology. These statements are not correct. The technique that has caused the significant decrease in the mortality rate is the Papanicolaou preparation and staining of smears which has been in use since the 1950-ies and had an impact since the 1970-ies. The liquid based preparation techniques is an improvement to the slide preparation that was proposed in the early 1990-ies and found widespread use several years later. Still very many cervical cytology screenings around the world are based on smears, not on liquid based preparations.

At line 146 you claim: "The TBS standard is not implemented in any AI system yet due to the complexity of subtypes, data availability, highly unbalanced classes, etc." That is not true. There is at least one study showing results for AI based classification according to the Bethesda standard: A More Comprehensive Cervical Cell Classification Using Convolutional Neural Network. Martin et.al. J of Am. Soc. of Cytopathology. Vol. 7, 2018. The paper is very short and the accuracy of that system is not impressive, still it shows previous work on this topic.

In the description of cervical cancer screening around line 100 you describe the lack of cytopathologists as the bottleneck in the screening. And it is true that the shortage of pathologists is a problem in many countries. But in almost all western countries the screening is not done by

cytopathologists but by cytotechnologists, specially trained professionals who do the primary screening without being medical doctors or pathologists. The pathologists only review positive cases before assigning final diagnosis. So it is cytotechnologists that are being replaced or supported by AI based screening methods. Is that not the case in China? Manpower is still a problem because there is also a lack of cytotechnologists in most countries.

In the material and methods section it is described that the slides were prepared with three different segmentation methods, line 164. I assume you mean sedimentation methods. Please correct.

But even after that correction it is unclear exactly what these methods are. There are a couple of widely used commercial methods for preparing liquid based PAP-smears. Are you using any of those please give a reference. If not give references to the methods you are using or provide a description at least in the supplemental material. Also provide reference describing what Ea36 and Ea50 are.

On line 166 it is stated that the images were classified according to figure 1A. That figure is full of short acronyms that are only explained in one of the tables in the supplemental material or not at all. The same holds for figure 1B. Most of them are related to the Bethesda standard, but the figure legend should still include an understandable explanation of what the acronyms stand for or at least a reference to the table where the explanation can be found.

On line 175 it is stated that the smears were scanned using a 40X scanner. But that does not define the digital image resolution. The relevant number is how big the pixels are in the object plane, and what effective numerical aperture the scanner used (determining whether the pixel size really corresponds to resolved optical resolution.)

On line 181 you say that you applied an optimized threshold. There are many ways of optimizing thresholds, please describe how you did it. Was it a fixed global threshold or varying, defined for each field?

Line 184, in your DPIQC system you are claiming high accuracy, but how was the ground truth determined. There is a rather fuzzy border between acceptable subjective image quality and non-acceptable. Was the decision about if an image was acceptable taken before the computer decision or after viewing the images after the classification, deciding if it was correct or not?

Around line 188 you are describing the challenging process of annotating the images. You say that three senior pathologists were reviewing positive smears. It is highly likely that there will be disagreements. How did you handle that? Did you use majority decision or require consensus. Did you remove smears where there were disagreements? There are many approaches to this problem. Please clarify. The same questions relates to the evaluation of the test results described at line 266. How were disagreements documented and resolved? How this is handled is important since it can have a strong influence on the final scores on accuracy etc.

On line 191 you say that the negative smears were reviewed by 20 pathologists. Does that mean that all the 50000 negative slides were looked at or was there some sampling? Did one pathologist look at each slide or did all 20 look at the selected slides? Again there are many ways of organizing this kind of challenging review. Please clarify.

On line 201 what does decomposed and labelled the lesion area according to the shape such that the diameter was within the given range mean? What diameter? Please clarify.

At line 235 you are saying that The highest weight of the verification set Mean Intersection Over Union (mIOU) in the five-fold cross validation training was selected as the final prediction weight. The table 01 contains a lot of numbers and percentages but no weights. How were the weights defined? It should also be clarified what the "number" in the table refer to (I am guessing it refers to the five folds).

At line 240 you say The scale of each type of image was roughly 1:1:1; the test data set was divided by 0.2 scale. What scales are you referring to here? Image magnification scale, relative number of images or what? Please clarify.

At line 258 you are saying that the final diagnostic decisions were obtained by "logical judgement" but you do not define how this logical judgement was achieved. For the results to be reproducible you need to describe the logical decision functions used.

At line 263 you write: using our developed online or local cervical liquid-based thin-layer cell smear AI auxiliary diagnostic system. Does that refer to the system described in the previous text

or are there different systems "local" versus "on-line" systems?

The Result section starting at line 268 contains a lot of methods details that I would have expected to find in the Methods section. There may be reasons for this organization of the paper but if it is retained I think there should be a comment early in the paper describing the logic behind the way it is organized.

At line 286 you mention detection speed for different CNN approaches and it is documented in Supp table 5. But you do not describe what is being detected. Is it the classification of one image field or one cell cluster or what? Please clarify in text and in table legend.

In general it is often hard to grasp if you are discussing performance on the image level or on the patient level. It seems that "samples" sometimes refers to image patches and sometimes to patient cases. Try to make that more clear. The importance of different types of errors vary a lot between the cellular or local image level and the patient level.

At line 370 you state Three cytopathologists confirmed all of the final diagnoses of positive test samples for intraepithelial lesions and infectious lesions. Was this confirmation done on the cytological specimens or were they clinically confirmed on histopathological examination of the tissue removed from the patients (I assume they had an operation performed at least for the patients with intraepithelial lesions, perhaps not for the infectious lesions). Since there is substantial variation between cytopathologist diagnoses based on cytological smears it is generally considered that the real ground truth is the histopathological diagnosis.

Figure 5 has a lot of category labels on the X-axes C1 ... C24 both in A and B these are not described, neither in the text nor in the figure legend. Also there are wavy lines at top at the graphs in addition to the bar diagram. There is no explanation what those lines represent. At the bottom of figure 5 there are a series of five large box-bars, all the same height but different colors. What does those represent?

In figure 6 the classification performance of the test set seems to be much better than for the training and validation set. That is usually not the case. Please explain better what these bars stand for. Also in the same figure in C you show bars for 10 different Weights but I can't find any explanation what those weights are.

In Supp. Figure 3 C you show feature importance for a large number of different features but I can nowhere find a description of what those features are. The features should be described at least in the supplementary material.

In line 313 you state that you reached 83.6% accuracy "as shown in figure 6A". But figure 6A does not show any percentages at all. Please clarify.

Finally at line 517 the authors agree to publish this in NEW ENGLAND JOURNAL OF MEDICINE once accepted. But this review is for Nature Communication?

Reviewed by Ewert Bengtsson

Reviewer #3:

Remarks to the Author:

Rapid TBS classification and diagnosis of cervical liquid-based thin-layer cell smears by AI-driven model based assistive diagnostic platform by Zhu et al is a well written paper regarding the development and testing of a deep learning algorithm applied to the diagnosis of ThinPrep™ cervical cytology. Drawing from a large retrospective set of samples and through the use of YOLOv3, the authors were able to generate a large number of annotated and classified images whilst minimizing human intervention in this cumbersome step. They also developed a quality control model, able to discard poor quality smears or those of inadequate cellularity. Then, using open-source frameworks, libraries and algorithms, implemented in a clever, intertwined fashion, the authors were able to create an AI classification system, which was able to classify WSI of different sample types into the different TBS categories with high sensitivity and specificity not only for the detection of intraepithelial squamous lesions but also microorganisms, such as candida or actinomyces. The classification system was then validated prospectively in a clinical trial. This paper, however, still has some limitations that need to be addressed:

1. The use of the English language in the paper is not perfect and needs to be improved significantly before publication; we suggest a professional review and correction by a native English speaker. Below are a few examples (please note this list is not extensive):
 - a. Page 4 Line 66 - consider rephrasing this sentence. A suggestion: "We achieved a sensitivity of 92% in the detection on squamous intraepithelial lesions and of 85% in the detection of infectious lesions and endometrial cells. Both values were superior to those of expert cytopathologists, yet much faster (<180 seconds)."
 - b. Page 5 Line 90 - Consider changing "The unique trend" to "This unique trend"
 - c. Page 5 Line 95 - Consider replacing more with better
 - d. Page 5 Line 98 - Consider removing "Technically"
 - e. Page 5 Line 100 - Consider removing "Additionally"
 - f. Page 5 Line 102 - Consider changing this paragraph. The conclusion ("Thus, the development of an Artificial Intelligence (AI) assistive diagnosis system will accelerate the work of cytopathologists, reduce the false negative rate, increase the diagnostic accuracy and ultimately reduce mortality.") is not entirely supported by the premises
 - g. Page 7 Line 142 - Consider changing the acronym "TIS", as it is used elsewhere in the literature to refer to "The International System for Serous Effusion Cytology"
 - h. Page 8 Line 162 - Replace "slices" with "slides"
 - i. Page 25 Line 517 - Change the journal to the appropriate one
 - j. Figure 3 - Change Ture for True - Review figures for typos
2. Some inaccuracies should be corrected
 - a. Page 5 Line 85 - cervical screening projects are not based only on Thinprep
3. There is a lack of references to other relevant publications using machine learning in cytopathology, namely using cervical cytology samples. References used are old. This is a major weakness of this publication - although a review of current literature is outside the scope of this paper, it would be very interesting to see how the methodology and results of the authors compare with other publications. A few examples of relevant, recent, publications that are worth of mention:
 - a. Bao, Heling, et al. "Artificial intelligence-assisted cytology for detection of cervical intraepithelial neoplasia or invasive cancer: A multicenter, clinical-based, observational study." *Gynecologic Oncology* 159.1 (2020): 171-178.
 - b. Bao, Heling, et al. "The artificial intelligence-assisted cytology diagnostic system in large-scale cervical cancer screening: A population-based cohort study of 0.7 million women." *Cancer medicine* 9.18 (2020): 6896-6906.
 - c. Tan, Xiangyu, et al. "Automatic model for cervical cancer screening based on convolutional neural network: a retrospective, multicohort, multicenter study." *Cancer Cell International* 21.1 (2021): 1-10.
 - d. Martínez-Más, José, et al. "Classifying Papanicolaou cervical smears through a cell merger approach by deep learning technique." *Expert Systems with Applications* 160 (2020): 113707.
4. Further details should be added regarding the preparation and digitizing of the slides:
 - a. How were staining protocols homogenized between different institutions? What hardware was used?
 - b. The authors do not specify the hardware used to digitize the slides. Was it a single digitized or multiple? Was Z-stacking used? If not, do you see that as a limitation?
5. The authors fail to discuss how they arrived at their methodology and the algorithms used. A technical, detailed explanation, as well as a brief summary of alternative solutions available would be appreciated (perhaps as a table).
6. It is unclear how the AI system was implemented in the workflow of the laboratory
 - a. In the prospective trials, did pathologists interact with the system? If not, how would you implement this system in a routine clinical practice?
 - b. Although the speed of the system is mentioned, there is no data on the overall impact on response times. Did the pathologists interact with the system at all during the prospective trials or were they blinded?
 - c. In the prospective trials, were all slides reviewed by the three senior cytopathologists? You calculate cytopathologist sensitivity and specificity, comparing with the AI. What is your gold

standard?

7. It would be interesting to see how the model compares to cytotechnicians in screening samples. It would also be interesting to see how the performance of cytopathologists changes when they are assisted by the AI system.

REPLY TO REVIEWER COMMENTS

Reviewer #1 (Remarks to the Author):

AI is a promising technology that could improve the diagnosis workflow in real clinical practice. This paper presented a computerized method to automatically process and classify digitalized TCTs based on the Bethesda system. My major critiques are:

1. In the 5-fold cross validation, the XGBoost model did not perform well in distinguishing ASCH, ASCUS, HSIL, LSIL, and SCC. For instance, LSIL had about 65% of accuracy. HSIL obtained the accuracy from about 66% to 73%. These values of accuracy also had a large variation. This might suggest that the model could not work properly with a low stability. The model robustness needs to be evaluated further.

Response: In cervical liquid-based cytology, the accurate classification and diagnosis of squamous intraepithelial lesions is difficult, as implied by the name of the squamous intraepithelial lesions defined in the Bethesda system (TBS): atypical squamous cells of undetermined significance (ASCUS) and atypical squamous cells--cannot exclude HSIL (ASCH). Since the morphology of squamous intraepithelial cells will gradually change with lesion progression, similar lesion morphologies may appear in different TBS classifications. For instance, ASCUS may be a serious lesion even though it is only "atypical cell," and it is hard to determine if the morphology of ASCUS is abnormal enough to meet the criteria of LSIL diagnosis. In addition, the morphology of ASCH and HSIL is almost the same; the differentiation between ASCH and HSIL is highly dependent on doctors' clinical experience most of the time. The above-mentioned factors may degrade the classification accuracy. The cytopathological diagnosis has huge variations too. Moreover, our current dataset is unbalanced, which can also lead to a higher misclassification rate.

We have included some new training data into the database and optimized our XGBoost model based on this new data. We performed 10-fold cross validation as you suggested, and the accuracy of the new model has improved (new 80.43% Vs old 75.32%, **Supp. Table 07**). However, for the categories having fewer samples (such as SCC and ASCH and other rare cases), the accuracy variation remain large. We have also applied the new model to prospective sample classification, and a very slightly improvement of accuracy was observed (New 74.15% Vs old 73.91%, **Supp. Table 15**). Based on the results of our prospective

studies, the average human diagnostic accuracy in actual clinical environments is 80.16% (**Supp. Table 17**), which is still not high enough for direct clinical application. However, this errors does not affect clinical treatment. Because patients with positive results (ASCUS, LSIL, ASCH, HSIL and SCC) need further cervical biopsy or HPV (human papilloma virus) testing, and only cervical biopsy results can ultimately guide the clinical treatment plan. In fact, minimizing the impact of two uncertain diagnoses ASCUS and ASCH in cervical liquid-based cytology diagnosis, thereby obtaining a more deterministic and accurate diagnosis to guide clinical treatment, is the focus of the next stage of our research. Of course, we need to collect more data (more samples to improve sample balance and follow-up of histological results) to improve the model. In the future, we believe that the accuracy of AI will probably exceed that of cytopathologists in TBS classification, closer to the results of cervical histological follow-up.

2. Why was the 5-fold across validation used? In general, 10-fold across validation is commonly adopted. By the other hand, the collected dataset is large enough to split into training and test dataset. The model could be well tested on an independent test dataset.

Response: Thanks for your valuable suggestion. It is true that 10-fold validation is optimal if the overall sample size is large. However, the rational of using 5-fold cross validation is mainly to account for rare cases as it is difficult to balance the population of rare cases with a larger number of folds. As mentioned in the previous question, we have performed 10-fold cross validation as you suggested and balanced the classes in different folds as much as we can. The accuracy of the new model has slightly improved compared to the old one, as described in **point #1**.

3. How large the variations between the data acquired from 11 hospitals? Did the variations degrade the model's performance?

Response: We have made statistics on the diagnosis results of 11 medical institutions. See (**Supp. Table 14**) for details. From the results, we can see that there are still certain differences between different medical centers. Whether retrospective samples are provided is also an important influencing factor, though without statistical significance (manuscript **line 432-437**).

4. Since multi-center data cohorts are used in this paper, it is a better way to build a model, which is trained on one cohort and tested on another one. Therefore, the model is well validated on independent dataset.

Response: We very much agree with your suggestion. There were eleven institutions in the sample collection for the prospective study. Except for five institutions that have provided the retrospective samples, the samples of the remaining six institutions are independent of the training set in the retrospective study. The results show that the sensitivity and specificity of the system in prospective tests of institutions that provided retrospective training samples were slightly higher than those of institutions that did not provide retrospective training samples (not statistically significant) (manuscript **line 432-437 and Supp Table 14**).

5. The AI-based classification could be compared with the human cytopathologists in terms of classification accuracy, sensitivity, and specificity.

Response: We have performed double-blind tests between senior cytologists and the AI system. The system and senior cytologists were asked to diagnose prospective samples based on TBS standards, and their sensitivity and specificity were recorded and compared. The results showed that the sensitivity of the system is higher than that of senior pathologist, while the specificity of the system is slightly lower than that of senior cytologists (**Figure 6D**).

6. After nuclear segmentation from the U-Net model, a total of 121 features were extracted from the segmented nuclei. Did these features match the TBS standard, in which morphological features are usually reviewed by the pathologists? What was the feature selection method used in the presented pipeline? Did this feature selection method work well to identify the important features? Could these selected features be useful to assist the pathologists during the diagnosis procedure?

Response: Sorry, we did not describe clearly. In fact, 121 features come from four deep learning models (YOLOv3, Xception, patch area classification and cell nucleus segmentation model). Some of the 121 features are experience features based on cytopathologists' domain knowledge according to the TBS standard, and some of them are determined by computation, i.e. computational features. They are not from the cell nucleus segmentation model alone, but

rather the diverse output from these multiple models. We have added the specific features table (**Supp Table 04**). In addition, we obtained feature importance distribution histograms (**Supp. Figure 2B**), this result showed that the importance of those selected features was related to the pathological characteristics of the lesions and will be helpful to provide feedback to the cytopathologists and enhance their performance.

7.The current quantitative results could not sufficiently support the authors' conclusion without more experiments and systematic analysis.

Response: As we mentioned earlier, we agree that additional experiments are necessary, and forthcoming, before direct application of our model to the clinic.

In this study, we collected a large number of retrospective samples and deployed multiple deep learning models to perform object detection and classification as well as accurate quantification of relative size (area) of the diseased nuclei. We combined the high-quality parameters extracted from the above models with the domain knowledge of cervical fluid-based cytology to develop a cervical liquid-based thin-layer cytology TBS classification system, including image quality control, squamous intraepithelial lesion classification as well as other lesion classification functions. As a prospective study, our system was highly consistent with corresponding histological biopsy results (**Table 02**), and its sensitivity exceeded that of senior cytologists (**Figure 6D**). Our system can adapt to different smear preparations as well as staining and scanning protocols and showed promising generalizability as seen in **Figure 6A-6C**.

Reviewer #2 (Remarks to the Author):

This manuscript describes a system developed to classify liquid based cell smears for early detection of cervical cancer according to the Bethesda system. The major contribution from this study is that they address the whole screening problem from handling specimens prepared at different labs through automatically finding and removing poorly prepared specimens, finding relevant cell clusters and classifying those based on general appearance through trained neural networks but also through segmentation of nuclei and extraction of features that can be used for refining the classification finally confirming the developed

system on a completely separate dataset. Very many papers are published addressing some of the problems e.g. segmenting or classifying cells but very few studies addresses the whole automated screening problem. Also most studies deal with a few hundred samples at most. A really strong feature is that this study is based on a massive number of cases, >81 000 for the training and >34 000 in the test set. The results show performance on the same level as human cytopathologists for the same material. The study is thus an important contribution to the state of the art in the field. It does however suffer from a number of issues that need to be addressed before publication.

Response: Thanks for your comments.

My most serious criticism of the paper is that the authors seem to have a rather vague idea about the history and state of the art in their field. This is particularly clear from the literature review in the introduction.

Response: We sincerely accept your criticism; we have addressed the concerns and corrected the literature review carefully. The relevant papers and reviews of the predecessors you mentioned have been carefully cited and marked in the **Reference section** of the manuscript in red font.

At line 138 it is correctly stated: Cervical cytology is an early application of computer-aided diagnosis and has applications to a wide range of cancer pathologies. But only one reference (reference 20) to one particular study in the field is given. This does in no way acknowledge the fact that image analysis assisted cervical cancer screening is the most studied of all applications in cytology and histopathology, probably in medicine as a whole. The first studies were done in a major US project in the 1950-ies see “W. E. Tolles and R. C. Bostrom, “Automatic screening of cytological smears for cancer: the instrumentation”, Annals of the New York Academy of Sciences, vol. 63, 1956.” Since then many thousand papers have been published. There were commercial systems for the screening available from the 1990-ies. One of many review articles describing the history of the field is “Screening for Cervical Cancer Using Automated Analysis of PAP-Smears.” Bengtsson and Malm, Computational and Mathematical Methods in Medicine”, 2014.

This lack of meaningful referencing of the literature in the field is also apparent in the description of the more recent work where AI technology has been applied to this problem.

The references describing the state of the art in deep learning applied to pathology or cytology diagnosis seem to be rather randomly chosen. Not particularly focused on the kind of applications the manuscript is about.

Response: We agree with your criticism. In the cervical cancer screen, there are indeed an abundance of references. As our paper focused on TBS classification of cervical liquid-based cytology, we included key relevant references in our manuscript to better reflect the development on this topic. We have carefully cited and marked the articles in the **Reference section** of the manuscript in **red font**.

Reference 12 published in 2009 does not as stated describe “AI solutions were also used in cytological screening, quantitative morphological analysis, histopathological diagnosis and prognosis prediction” since it was published in 2009, before the modern boom in AI deep learning technology started. There were work on neural networks applied to this problem far before the current boom see e.g. L. J. Mango, “Computer-assisted cervical cancer screening using neural networks,” Cancer Letters, vol. 77, 1994, but they were not discussed in reference 12 and due to the limited computational power available at that time they were not as successful as more traditional image analysis approaches.

Response: As above, we cited the wrong literature and have modified the original manuscript.

A more recent review that does address also AI based approaches is: Image analysis and machine learning in digital pathology: Challenges and opportunities. Med Image Anal. 2016. An even more recent and more comprehensive review is: Deep learning in image cytometry: a review. A Gupta et.al. Cytometry Part A, 2019, it includes papers on cervical cytology screening. Another recent and highly relevant review is: Artificial intelligence in cytopathology: a review of the literature and overview of commercial landscape. Landau and Pantananowitz. Journal of the American Society of Cytopathology, Vol 8, 2019. Even after the publication of those reviews a number of highly relevant papers have been published in the very active field of deep learning applied to cervical cytology screening.

Response: Thank you for your suggestion. They are indeed closely relate to artificial intelligence of cervical cancer screening. We have cited these papers in our manuscript.

There are also a number of other strange statements in the introductory literature review where the titles do not seem to fit the topic of their citations. For instance reference 13 is said to deal with lung cancer although the actual topic is colorectal cancer. So the whole description of the field and the literature review need to be significantly updated.

Response: Sorry, we made a mistake when inserting the literature and have modified the original manuscript.

At Line 84 it is stated: This global success is mainly attributed to the establishment of cervical cancer screening projects based on the Thinprep Cytologic Test (TCT) technology. TCT has laid a solid foundation for early diagnosis followed clinical treatment. Similarly at line 404 you state: Since the 1970s, the mortality rate caused by cervical cancer in many countries has been on an overall downward trend, which is largely due to the development of cervical liquid-based thin-layer cell smear technology. These statements are not correct. The technique that has caused the significant decrease in the mortality rate is the Papanicolaou preparation and staining of smears which has been in use since the 1950-ies and had an impact since the 1970-ies. The liquid based preparation techniques is an improvement to the slide preparation that was proposed in the early 1990-ies and found widespread use several years later. Still very many cervical cytology screenings around the world are based on smears, not on liquid based preparations.

Response: Thank you for your suggestion. We have modified the corresponding statement in the manuscript (**line 103-108 and line 485-489**). You're right. In fact, traditional Pap stain smears have been gradually replaced by cervical liquid-based cell smears in China and many countries because of the high false-negative rate.

At line 146 you claim: "The TBS standard is not implemented in any AI system yet due to the complexity of subtypes, data availability, highly unbalanced classes, etc." That is not true. There is at least one study showing results for AI based classification according to the Bethesda standard: A More Comprehensive Cervical Cell Classification Using Convolutional Neural Network. Martin et.al. J of Am. Soc: of Cytopathology. Vol. 7, 2018. The paper is very short and the accuracy of that system is not impressive, still it shows previous work on this topic.

In the description of cervical cancer screening around line 100 you describe the lack of cytopathologists as the bottleneck in the screening. And it is true that the shortage of pathologists is a problem in many countries. But in almost all western countries the screening is not done by cytopathologists but by cytotechnologists, specially trained professionals who do the primary screening without being medical doctors or pathologists. The pathologists only review positive cases before assigning final diagnosis. So it is cytotechnologists that are being replaced or supported by AI based screening methods. Is that not the case in China? Manpower is still a problem because there is also a lack of cytotechnologists in most countries.

In the material and methods section it is described that the slides were prepared with three different segmentation methods, line 164. I assume you mean sedimentation methods. Please correct.

Response: Thanks for your advice on the related studies. The AI system we refer to is an applicable medical solution. We have cited relevant literature in the manuscript (**reference 29**), and we appreciate the efforts of our predecessors. As you alluded to, the study you mentioned here is not comprehensive, only classifying some specific categories in the diagnosis of TBS. Besides, it mainly focuses on TBS classification, without any automatic detection of diagnostic clues or automatic diagnosis. This article performs TBS classification, which is similar to the Xception or Patch classification model in our research. We have thus extended prior efforts by building a comprehensive TBS classification and automated lesion detection and diagnosis system.

Yes you are right. China has a different healthcare system. In China, currently only cytologists conduct screening, almost without assistance from cytotechnologists, while cytopathologists are responsible for re-examination. Thus, our method would ameliorate the urgent need for cytopathologists in China and those performing the role of cytotechnologists more broadly.

The typo of “Segmentation” is wrong, we have corrected this error. Thanks for pointing it out.

But even after that correction it is unclear exactly what these methods are. There are a couple of widely used commercial methods for preparing liquid based PAP-smears. Are you using any of those please give a reference. If not give references to the methods you are using or

provide a description at least in the supplemental material. Also provide reference describing what Ea36 and Ea50 are.

Response: Thanks for your advice. We have provided three smear preparation methods in the manuscript and introduced the Pap staining scheme. For details, see the line 221-228.

On line 166 it is stated that the images were classified according to figure 1A. That figure is full of short acronyms that are only explained in one of the tables in the supplemental material or not at all. The same holds for figure 1B. Most of them are related to the Bethesda standard, but the figure legend should still include an understandable explanation of what the acronyms stand for or at least a reference to the table where the explanation can be found.

Response: Thanks for your advice. We have improved the data presentation you mentioned in the figures, legends, and table. Please refer to the new Figures 2A and Table 01. In our manuscript we also provided clear explanations of the acronyms in new Figure legends 3B and 3C.

On line 175 it is stated that the smears were scanned using a 40X scanner. But that does not define the digital image resolution. The relevant number is how big the pixels are in the object plane, and what effective numerical aperture the scanner used (determining whether the pixel size really corresponds to resolved optical resolution.)

Response: Thank you for your advice. We have corrected the manuscript (line 552-554) and uniformly use 0.25um/pixel as the resolution for the scanned images.

On line 181 you say that you applied an optimized threshold. There are many ways of optimizing thresholds, please describe how you did it. Was it a fixed global threshold or varying, defined for each field?

Response: Thank you for your correction. We used the Otsu threshold segmentation method and have corrected it in the manuscript (line 718-720).

Line 184, in your DPIQC system you are claiming high accuracy, but how was the ground truth determined. There is a rather fuzzy border between acceptable subjective image quality

and non-acceptable. Was the decision about if an image was acceptable taken before the computer decision or after viewing the images after the classification, deciding if it was correct or not?

Response: Yes, you are right. In many cases, cytopathologists do have difficulty defining the fuzzy boundaries in judging whether the smear is satisfactory for diagnosis. The training data for DPIQC system, which contains both satisfying/unsatisfying images, was therefore carefully selected and confirmed by three cytopathologists together, that is, human decision is the gold standard for DPIQC system (**line 710-712**). For artificial intelligence assisted systems, in addition to whether the production and staining quality, the quality of scanning is also an important influence factor for diagnosis accuracy. Therefore, taking into account the image quality requirements of the artificial intelligence assistance system, our system will give special prompts for unsatisfactory samples. For those unsatisfactory smears, the cytopathologist needs to make his or her own judgment as final decision. We compared the performance of DPIQC system and human, 89.11% of the unsatisfactory samples predicted by the DPIQC system are consistent with the final decision of the cytopathologists. The data and details are presented in **Supp. Table 12**.

Around line 188 you are describing the challenging process of annotating the images. You say that three senior pathologists were reviewing positive smears. It is highly likely that there will be disagreements. How did you handle that? Did you use majority decision or require consensus. Did you remove smears where there were disagreements? There are many approaches to this problem. Please clarify. The same questions relates to the evaluation of the test results described at line 266. How were disagreements documented and resolved? How this is handled is important since it can have a strong influence on the final scores on accuracy etc.

Response: The positive (except for NILM) smears were reviewed by three cytopathologists who diagnosed according to the TBS standard. First, they referred to the histological diagnosis of the cervical biopsy corresponding to the smear. For smears that had no histological results or false-negative histological results which might be caused by irregular cervical biopsies, the unanimous diagnosis of the three cytopathologists shall prevail. For smears that could not reach a unanimous diagnosis, the positive diagnosis with the highest lesion grade shall prevail (**line 554-561 and line 730-731**).

On line 191 you say that the negative smears were reviewed by 20 pathologists. Does that mean that all the 50000 negative slides were looked at or was there some sampling? Did one pathologist look at each slide or did all 20 look at the selected slides? Again there are many ways of organizing this kind of challenging review. Please clarify.

Response: The negative (NILM) smears were reviewed by senior cytologists. Negative smears were evenly distributed to 20 senior cytologists for re-examination, and three cytopathologists randomly selected 10% of the smears for spot check to ensure that there were no positive cases in negative smears (**line 561-565**).

On line 201 what does decomposed and labelled the lesion area according to the shape such that the diameter was within the given range mean? What diameter? Please clarify.

Response: We are sorry that we have not described it clearly in the manuscript. Since the size of the annotated image was limited to 608 pixels by 608 pixels, for any image whose width or height exceeded 608 pixels, the cytologists decomposed the image into several small images according to the morphological structure of the lesion area such that the width or height of the decomposed images were within 608 pixels (**Supp. Figure 4B**). We have clarified the manuscript to reflect this description (**lines 577-580**).

At line 235 you are saying that The highest weight of the verification set Mean Intersection Over Union (mIOU) in the five-fold cross validation training was selected as the final prediction weight. The table 01 contains a lot of numbers and percentages but no weights. How were the weights defined? It should also be clarified what the “number” in the table refer to (I am guessing it refers to the five folds).

Response: Sorry for our mistake, What you mentioned is our patch classification model. “Weight” actually refers to “model”, we are now using 10-fold cross-validation. The “model” (not “weight”) with the highest accuracy in the 10-fold cross-validation training was selected as the final prediction model. Please see our revised manuscript for the detailed method (**line 619-621**). The Patch model results of training, verification and testing are shown in the **Figure 3D**.

At line 240 you say The scale of each type of image was roughly 1:1:1; the test data set was divided by 0.2 scale. What scales are you referring to here? Image magnification scale, relative number of images or what? Please clarify.

Response: We have made changes in revised manuscript “ The quantity ratio of each type of image was roughly 1:1:1, 80% of the selected dataset was used for training while 20% for validation.”, see **line 625-626**.

At line 258 you are saying that the final diagnostic decisions were obtained by “logical judgement” but you do not define how this logical judgement was achieved. For the results to be reproducible you need to describe the logical decision functions used.

Response: We have modified and explained in detail in the manuscript, see line???? . We have renamed logical judgment to Logical Decision Tree (Given in **Supp, Table 08**).

At line 263 you write: using our developed online or local cervical liquid-based thin-layer cell smear AI auxiliary diagnostic system. Does that refer to the system described in the previous text or are there different systems “local” versus “on-line” systems?

Response: We have two ways to deploy the system to the medical centers according to their preferences and IT regulatory policies. One is through an online system (server located at our side) and another is a dedicated physical server located inside the medical centers. Both of them are running the same model, i.e., our AIATBS system. We have modified and explained in detail in the manuscript, see **line 724-726**.

The Result section starting at line 268 contains a lot of methods details that I would have expected to find in the Methods section. There may be reasons for this organization of the paper but if it is retained I think there should be a comment early in the paper describing the logic behind the way it is organized.

Response: Thank you for your suggestion, we have improved the organization of the manuscript. We hope it is much easier to follow and understand.

At line 286 you mention detection speed for different CNN approaches and it is documented in Supp table 5. But you do not describe what is being detected. Is it the classification of one image field or one cell cluster or what? Please clarify in text and in table legend.

Response: YOLOv3 and several algorithms were evaluated to detect squamous intraepithelial lesions in this study. The old Supp. table 5 is a comparison result in order to screen a suitable detection algorithm to build our system. Sorry for our unclear description, we have modified the manuscript and explained in the legend of the table, see **line 250-256 and Supp. Table 02.**

In general it is often hard to grasp if you are discussing performance on the image level or on the patient level. It seems that “samples” sometimes refers to image patches and sometimes to patient cases. Try to make that more clear. The importance of different types of errors vary a lot between the cellular or local image level and the patient level.

Response: Thank you very much for your suggestion. We have carefully reviewed the manuscript to avoid confusion.

At line 370 you state Three cytopathologists confirmed all of the final diagnoses of positive test samples for intraepithelial lesions and infectious lesions. Was this confirmation done on the cytological specimens or were they clinically confirmed on histopathological examination of the tissue removed from the patients (I assume they had an operation performed at least for the patients with intraepithelial lesions, perhaps not for the infectious lesions). Since there is substantial variation between cytopathologist diagnoses based on cytological smears it is generally considered that the real ground truth is the histopathological diagnosis.

Response: As you mentioned earlier about the diagnostic criteria, here we are still using the previous criteria, that is, “the positive (except for NILM) smears were reviewed by three cytopathologists who diagnosed according to the TBS standard. First, they referred to the histological diagnosis of the cervical biopsy corresponding to the smear. For smears that had no histological results or false-negative histological results which might be caused by irregular cervical biopsies, the unanimous diagnosis of the three cytopathologists shall prevail. For smears that could not reach a unanimous diagnosis, the positive diagnosis with the highest lesion grade shall prevail” (**line 554-561**). For comparison, we also collected 980

cervical biopsy results of prospective samples from three large medical institutions. We found that the sensitivity of the AI-assisted diagnosis system for cervical intraepithelial lesions was 94.74% (**Table 02**).

Figure 5 has a lot of category labels on the X-axes C1 ... C24 both in A and B these are not described, neither in the text nor in the figure legend. Also there are wavy lines at top at the graphs in addition to the bar diagram. There is no explanation what those lines represent. At the bottom of figure 5 there are a series of five large box-bars, all the same height but different colors. What does those represent?

Response: Thank you for your reminder. We have replaced the old Figure 5 A and B with the **new Figure 3B and 3C**, and provided further explanations in the **new figure legends of Figure 3B and 3C**. The old Figure 5C,5D and 5E have been replaced by the **new Figure 3D**.

In figure 6 the classification performance of the test set seems to be much better than for the training and validation set. That is usually not the case. Please explain better what these bars stand for. Also in the same figure in C you show bars for 10 different Weights but I can't find any explanation what those weights are.

Response: Sorry, we made a mistake in the original Figure 6A. We are showing the effect of the cell nucleus segmentation model (**new Figure 3E**). The accuracy of the cross-validation corresponding to this model is shown in **Supp Table 03**. For old Figure 6C, we have explained in the **figure legend of Supp. Figure 2C**.

In Supp. Figure 3 C you show feature importance for a large number of different features but I can nowhere find a description of what those features are. The features should be described at least in the supplementary material.

Response: Thank you for your reminder, the details have been provided in the **Supp. Table 04**.

In line 313 you state that you reached 83.6% accuracy “as shown in figure 6A”. But figure 6A does not show any percentages at all. Please clarify.

Response: Thank you for your reminder. This mistake is related to our correction above. In old Figure 6A, we needed to show the effect of the cell nucleus segmentation model, which we have corrected in the updated figure (**Figure 3E**). The accuracy of the cross-validation corresponding to this model is shown in **Supp Table 03**.

Finally at line 517 the authors agree to publish this in NEW ENGLAND JOURNAL OF MEDICINE once accepted. But this review is for Nature Communication?

Response: Sorry for the mistake. This work was previously submitted to NEJM. We have corrected the error (**line 900**).

Reviewed by Ewert Bengtsson

Reviewer #3 (Remarks to the Author):

Rapid TBS classification and diagnosis of cervical liquid-based thin-layer cell smears by AI-driven model based assistive diagnostic platform by Zhu et al is a well written paper regarding the development and testing of a deep learning algorithm applied to the diagnosis of ThinPrep™ cervical cytology. Drawing from a large retrospective set of samples and through the use of YOLOv3, the authors were able to generate a large number of annotated and classified images whilst minimizing human intervention in this cumbersome step. They also developed a quality control model, able to discard poor quality smears or those of inadequate cellularity. Then, using open-source frameworks, libraries and algorithms, implemented in a clever, intertwined fashion, the authors were able to create an AI classification system, which was able to classify WSI of different sample types into the different TBS categories with high sensitivity and specificity not only for the detection of intraepithelial squamous lesions but also microorganisms, such as candida or actinomyces. The classification system was then validated prospectively in a clinical trial.

This paper, however, still has some limitations that need to be addressed:

1. The use of the English language in the paper is not perfect and needs to be improved significantly before publication; we suggest a professional review and correction by a native English speaker. Below are a few examples (please note this list is not extensive):

- a. Page 4 Line 66 - consider rephrasing this sentence. A suggestion: “We achieved a sensitivity of 92% in the detection on squamous intraepithelial lesions and of 85% in the detection of infectious lesions and endometrial cells. Both values were superior to those of expert cytopathologists, yet much faster (<180 seconds).”
- b. Page 5 Line 90 – Consider changing “The unique trend” to “This unique trend”
- c. Page 5 Line 95 – Consider replacing more with better
- d. Page 5 Line 98 – Consider removing “Technically”
- e. Page 5 Line 100 – Consider removing “Additionally”
- f. Page 5 Line 102 – Consider changing this paragraph. The conclusion (“Thus, the development of an Artificial Intelligence (AI) assistive diagnosis system will accelerate the work of cytopathologists, reduce the false negative rate, increase the diagnostic accuracy and ultimately reduce mortality.”) is not entirely supported by the premises
- g. Page 7 Line 142 – Consider changing the acronym “TIS”, as it is used elsewhere in the literature to refer to “The International System for Serous Effusion Cytology”
- h. Page 8 Line 162 – Replace “slices” with “slides”
- i. Page 25 Line 517 – Change the journal to the appropriate one
- j. Figure 3 – Change Ture for True – Review figures for typos

Response: Thank you for your reminder. We have carefully made these modifications. The revision is extensive, it will be difficult to reply above point by point, while the language is substantially improved.

2. Some inaccuracies should be corrected

- a. Page 5 Line 85 – cervical screening projects are not based only on Thinprep

Response: You are right. In addition to cervical cytology smears, cervical cancer screening includes many other tests. We have corrected it.

3. There is a lack of references to other relevant publications using machine learning in cytopathology, namely using cervical cytology samples. References used are old. This is a major weakness of this publication – although a review of current literature is outside the scope of this paper, it would be very interesting to see how the methodology and results of the authors compare with other publications. A few examples of relevant, recent, publications that are worth of mention:

- a. Bao, Heling, et al. "Artificial intelligence-assisted cytology for detection of cervical intraepithelial neoplasia or invasive cancer: A multicenter, clinical-based, observational study." *Gynecologic Oncology* 159.1 (2020): 171-178.
- b. Bao, Heling, et al. "The artificial intelligence-assisted cytology diagnostic system in large-scale cervical cancer screening: A population-based cohort study of 0.7 million women." *Cancer medicine* 9.18 (2020): 6896-6906.
- c. Tan, Xiangyu, et al. "Automatic model for cervical cancer screening based on convolutional neural network: a retrospective, multicohort, multicenter study." *Cancer Cell International* 21.1 (2021): 1-10.
- d. Martínez-Más, José, et al. "Classifying Papanicolaou cervical smears through a cell merger approach by deep learning technique." *Expert Systems with Applications* 160 (2020): 113707.

Response: Thank you for your suggestion. We have cited this important studies from the predecessors you mentioned in our manuscript and marked them in the **Reference section** of the manuscript in **blue font**.

4. Further details should be added regarding the preparation and digitizing of the slides:
 - a. How were staining protocols homogenized between different institutions? What hardware was used?

Response: As the information provided in our manuscript, see **line 221-228**. Both natural sedimentation and membrane sedimentation smears preparation use fully automatic liquid-based thin-layer cell film makers, and centrifugal sedimentation smears use centrifugation and manual operation. Staining uses two different Pap staining solutions, Papanicolaou Stain, EA-50 Solution and Papanicolaou Stain, EA-36 Solution, so we cannot homogenize the different production and dyeing smears. Fortunately, we can eliminate the influence of these factors on system generalization by maintaining the diversity of training on smears processed by these different production and staining methods.

- b. The authors do not specify the hardware used to digitize the slides. Was it a single digitized or multiple? Was Z-stacking used? If not, do you see that as a limitation?

Response: Thank you for your suggestion. We have added the information of smear scanning in the manuscript, see **line 552-554**.

5. The authors fail to discuss how they arrived at their methodology and the algorithms used. A technical, detailed explanation, as well as a brief summary of alternative solutions available would be appreciated (perhaps as a table).

Response: It is not easy to achieve the classification according to the TBS system and the AIATBS system is integrated with a few models. We significantly improved the method section to clarify the rationale of the system design (**line 658-705**). In addition, the **Supp. Figure 03** is designed to explain the system processing pipeline. The intermediate results of the different models and system for training, validation and testing are all included in the Supp. figures and tables. We hope this significant improved of our manuscript will address the concern of reviewer.

6. It is unclear how the AI system was implemented in the workflow of the laboratory.

a. In the prospective trials, did pathologists interact with the system? If not, how would you implement this system in a routine clinical practice?

Response: In the prospective experiment, the senior cytologists and the system were double-blinded and did not interact with each other. The comparison is between the human performance and AIATBS performance. Regarding the assistive diagnosis daily clinical applications, we have another on-going large-scale study dedicated to address this point.

b. Although the speed of the system is mentioned, there is no data on the overall impact on response times.

Response: We have compared the diagnosis time of cytologists with and without the assistance from AI in another on-going large-scale study, the result is shown in **Table 2 of this document**. We have also made statistics on the analysis time of different smear preparation methods, staining and scanners. We found that the smear preparation method and scanner type significantly impact the analysis time, while the staining method has no significant effect (**Supp. Table 16**).

Did the pathologists interact with the system at all during the prospective trials or were they blinded?

Response: In the prospective experiments, the senior cytologists and the system were blinded to each other.

c. In the prospective trials, were all slides reviewed by the three senior cytopathologists? You calculate cytopathologist sensitivity and specificity, comparing with the AI. What is your gold standard?

Response: Thank you for your reminder, as stated in our revised manuscript **line 554-565 and line 730-731**. For prospective study, the protocol of determining the final diagnosis was the same as the retrospective study as described, that is, the positive (including intraepithelial lesions, infectious lesions and EMC) smears were reviewed by three cytopathologists who diagnosed according to the TBS standard. For intraepithelial lesion smear, they first referred to the histological diagnosis of the cervical biopsy corresponding to the smear. For smears that had no histological results or false-negative histological results which might be caused by irregular cervical biopsies, the unanimous diagnosis of the three cytopathologists shall prevail. For smears that could not reach a unanimous diagnosis, the positive diagnosis with the highest lesion grade shall prevail. The negative (NILM) smears were reviewed by senior cytologists. Negative smears were evenly distributed to 20 senior cytologists for re-examination, and three cytopathologists randomly selected 10% of the smears for spot check to ensure that there were no positive cases in negative smears.

7. It would be interesting to see how the model compares to cytotechnicians in screening samples. It would also be interesting to see how the performance of cytopathologists changes when they are assisted by the AI system.

Response: Our system has not yet received regulatory approval for clinical application, i.e., in-field assistive diagnosis. We are applying for a large-scale clinical trial involved 30 medical institutions in China. We conducted a pilot clinical research project including four of the institutions to measure the effectiveness of artificial intelligence systems in assisting senior cytologists. The collected results show that the AI assistive system can significantly improve the diagnostic sensitivity of senior cytologists and is 6 times faster compared to

manual (as in **Table 1 and 2 of this document**). The study will be published as another paper soon with the large-scale clinical validation study in the near future.

Table 1 Sensitivity and specificity of AI and AI-assisted cytologist for TBS report

TBS classifications	Number of final diagnosis	Number of AI diagnosis	AI Sensitivity	Number of AI-assisted cytologist diagnosis	AI-assisted cytologist Sensitivity
ASCUS	499	449	89.98%	495	99.20%
LSIL	130	129	99.23%	130	100.00%
ASCH	161	154	95.65%	160	99.38%
HSIL	51	51	100.00%	51	100.00%
SCC	6	6	100.00%	6	100.00%
AGC-NOS	57	52	91.23%	56	98.25%
AGC-FN	7	5	71.43%	7	100.00%
ADC	5	5	100.00%	5	100.00%
CC	426	398	93.43%	423	99.30%
TRI	102	73	71.57%	95	93.14%
CAN	303	228	75.25%	295	97.36%
ACTINO	28	25	89.29%	28	100.00%
HSV	7	6	85.71%	7	100.00%
EMC	40	27	67.50%	38	95.00%
NILM	14592	12698	87.02%	14460	99.10%

Table 2 Comparison of smears reading time in different reading ways for different production methods

Methods	AI	Cytologist	AI-assisted cytologist	Cytologist/AI-assisted cytologist
Natural sedimentation	78.2s	122.5s	20.9s	5.86
Centrifugal sedimentation	73.5s	125.2s	21.2s	5.91
Membrane sedimentation	148.3s	173.5s	27.4s	6.32
Average				6.03

Reviewers' Comments:

Reviewer #2:

Remarks to the Author:

The authors have addressed all my questions and comments in my review properly and I have no further questions. This version gives in my opinion a good presentation of the research and its results.

A minor detail: in line 306 and Figure 3 "Ture_HASIL" is discussed. You most likely mean "True_HASIL". The same error is made in one of the supplementary figures.

Reviewer #3:

Remarks to the Author:

Thank you for your thorough and well thought point-to-point response.

The additional tables you provided are informative, namely those pertaining to the performance of assisted diagnosis. It is interesting to see you are already preparing the next steps in this research project. The use of the English language has also been greatly improved, which we appreciate. We believe this paper to be a valuable contribution to the field of digital pathology and computer assisted diagnosis.

Some minor issues you might wish to improve:

Line 108 - Insert a space between for and investigation

Line 149 - Consider replacing reported with published

Line 182 - Consider changing " (intraepithelial lesions, infectious lesions and Endometrial cell)" to "(squamous, glandular and infectious lesions)"

Line 364 - "To" is repeated; "limited features are sufficient".

Line 559 and 561 - Consider changing "shall prevail" to "prevailed"

REPLY TO REVIEWER COMMENTS

Reviewer #2 (Remarks to the Author):

The authors have addressed all my questions and comments in my review properly and I have no further questions. This version gives in my opinion a good presentation of the research and its results.

A minor detail: in line 306 and Figure 3 "Ture_HSIL" is discussed. You most likely mean "True_HSIL". The same error is made in one of the supplementary figures.

Response: Thank you for pointing out our spelling mistakes. We have corrected the spelling errors in the **Figure 3 legend** and **Supp. Figure 3**. In addition, we have carefully checked the entire manuscript to ensure that there are no such errors.

Finally, we would like to thank you again for taking the time to review our manuscript.

Reviewer #3 (Remarks to the Author):

Thank you for your thorough and well thought point-to-point response.

The additional tables you provided are informative, namely those pertaining to the performance of assisted diagnosis. It is interesting to see you are already preparing the next steps in this research project. The use of the English language has also been greatly improved, which we appreciate. We believe this paper to be a valuable contribution to the field of digital pathology and computer assisted diagnosis.

Some minor issues you might wish to improve:

1. Line 108 - Insert a space between for and investigation

Response: Thank you very much. We have made the correction in the manuscript (**line 105**).

2. Line 149 - Consider replacing reported with published

Response: We have replaced it in the manuscript based on your suggestion (**line 139**).

3. Line 182 - Consider changing " (intraepithelial lesions, infectious lesions and Endometrial cell)" to "(squamous, glandular and infectious lesions)"

Response: This is a very good suggestion, we have made changes in the manuscript as suggested. **(line172)**.

4. Line 364 - "To" is repeated; "limited features are sufficient".

Response: Thank you for pointing out these two errors, we have corrected them in the manuscript **(line 319)**.

5. Line 559 and 561 - Consider changing "shall prevail" to "prevailed"

Response: Thanks for your suggestion, we have amend according your suggestion **(line 499 and 501)**.

Finally, we would like to thank you again for taking the time to review our manuscript.